# Electrical and Electrochemical Sensors Based on Carbon Nanotubes for the Monitoring of Chemicals in Water—A Review

**DOI:** 10.3390/s22010218

**Published:** 2021-12-29

**Authors:** Gookbin Cho, Sawsen Azzouzi, Gaël Zucchi, Bérengère Lebental

**Affiliations:** 1Laboratoire de Physique des Interfaces et des Couches Minces (LPICM), Centre National de la Recherche Scientifique (CNRS), Ecole Polytechnique, IP Paris, 91128 Palaiseau, France; gookbin.cho@polytechnique.edu (G.C.); sawsen.azzouzi@yahoo.fr (S.A.); gael.zucchi@polytechnique.edu (G.Z.); 2Laboratoire Instrumentation, Simulation et Informatique Scientifique (LISIS), Département Composants et Systèmes (COSYS), Université Gustave Eiffel, 77447 Marne-La-Vallée, France

**Keywords:** carbon nanotubes, nanomaterials, water quality, chemical sensor, chemistor, field effect transistor, electrochemical sensors

## Abstract

Carbon nanotubes (CNTs) combine high electrical conductivity with high surface area and chemical stability, which makes them very promising for chemical sensing. While water quality monitoring has particularly strong societal and environmental impacts, a lot of critical sensing needs remain unmet by commercial technologies. In the present review, we show across 20 water monitoring analytes and 90 references that carbon nanotube-based electrochemical sensors, chemistors and field-effect transistors (chemFET) can meet these needs. A set of 126 additional references provide context and supporting information. After introducing water quality monitoring challenges, the general operation and fabrication principles of CNT water quality sensors are summarized. They are sorted by target analytes (pH, micronutrients and metal ions, nitrogen, hardness, dissolved oxygen, disinfectants, sulfur and miscellaneous) and compared in terms of performances (limit of detection, sensitivity and detection range) and functionalization strategies. For each analyte, the references with best performances are discussed. Overall, the most frequently investigated analytes are H^+^ (pH) and lead (with 18% of references each), then cadmium (14%) and nitrite (11%). Micronutrients and toxic metals cover 40% of all references. Electrochemical sensors (73%) have been more investigated than chemistors (14%) or FETs (12%). Limits of detection in the ppt range have been reached, for instance Cu(II) detection with a liquid-gated chemFET using SWCNT functionalized with peptide-enhanced polyaniline or Pb(II) detection with stripping voltammetry using MWCNT functionalized with ionic liquid-dithizone based bucky-gel. The large majority of reports address functionalized CNTs (82%) instead of pristine or carboxyl-functionalized CNTs. For analytes where comparison is possible, FET-based and electrochemical transduction yield better performances than chemistors (Cu(II), Hg(II), Ca(II), H_2_O_2_); non-functionalized CNTs may yield better performances than functionalized ones (Zn(II), pH and chlorine).

## 1. Introduction

While fresh water represents 3% of the total water on Earth, only 0.01% is available for human consumption [1]. Rapid population growth, unsustainable water use in agriculture and industry and climate changes are bringing about hydric stress worldwide. While drink water availability decreases, its quality also degrades: World Health Organization (WHO) reports that, in developing countries, 80% of human diseases, are water borne [2]. Drinking water quality in numerous countries does not meet WHO standards [3,4]. The presence of water contaminants critically impacts human beings and ecosystem. It is thus of vital importance to be able to analyze fresh water, whether it is groundwater, irrigation water or tap water.

Concerning water networks, water quality monitoring and control mainly takes place at the water supply intake or at the water treatment plant. However, this seems inadequate given the important variations in water quality observed throughout water distribution systems [5]. Online sensing—also called on-site sensing—is currently seen as the best solution to provide continuous, early warning systems for chemical contamination throughout the water network (from drink water production to waste water treatment). It designates the capability to monitor water quality accurately and in real-time, and it is expected to yield public health improvements via improved water safety [6]. To cover recent advances in the field, Kruse recently reviewed chemical sensors for water quality evaluation [7]. After detailing the parameters and contaminants that are currently relevant to water monitoring, the authors present exhaustively transduction methods for water quality sensors. The review shows that, despite worldwide efforts, there are still plenty of challenges to be met by online water quality sensors: reduction of costs and calibration frequency, increase in sensitivity and selectivity, reduction of power consumption and size and enhancement of lifetime [8].

These challenges have motivated a wide range of studies toward water quality sensors based on nanomaterials, for instance as described in references [9,10,11], as nanomaterials-based sensors are well-known to meet those specific challenges across all fields of research on sensors [12,13]. Among sensors fabricated with nanomaterials, those comprising carbon nanotubes (CNT) have been continuously proposed for chemical sensing since the early days of CNT research [14] taking advantage of their excellent chemical stability and their large surface area. Most recently, Schroeder et al. [15] reviewed CNT-based chemical sensors, with applications covering gas sensors, biosensors, food sensors or aqueous sensors.

However, there has been no review focusing specifically on water quality monitoring based on carbon nanotube sensors. The present paper endeavors to fill this gap. More specifically, we report on CNT-based electrical and electrochemical sensors, because they are particularly well-suited for online water monitoring applications [16,17,18]. The electrical transduction options for CNT-based chemical sensors are electrochemical, resistive, field-effect-based and electromechanical. The latter has only been reported for gas sensing, so is not discussed in the present paper [19]. The various CNT-based optical sensors are also beyond the scope of this paper. While they have been often proposed for water quality monitoring (see reference [20] for instance), their applicative use requires integration into microfluidic lab-on-chips, whose specifications are not suitable today for online use and much more appropriate for sampling-based (portable or off-site) water monitoring systems [21].

After a brief overview of the design, fabrication and operating principles of CNT-based chemical sensors, we present exhaustively the various electrical and electrochemical sensors reported in the literature from 2000 to mid-2021. We sort them by types of analytes and the various reports are analyzed in terms of sensing performances. Selected papers are highlighted in view of understanding the sensing mechanisms. Finally, the sensing performances of CNT-based sensors are compared to those of other upcoming nanomaterials.

## 2. Operating Principles of CNT-Based Chemical Sensors

### 2.1. Chemical Nanosensors

In general, a chemical sensor transforms chemical information (typically the presence or concentration of a target analyte in water) into an exploitable electrical signal. It consists of a chemical recognition layer (receptor) and a physicochemical transducer. The receptor interacts with target analytes, which affects the transducer then turns it into an exploitable signal [22]. When either the transducer or the recognition layer contains a nanomaterial or is nanostructured, the device is said to be a nanosensor.

The performance of a chemical sensor is notably characterized by its response curve, namely the relationship linking the sensor signal to the analyte concentration. The response is more often modeled as linear, though exponential and logarithmic responses are also reported. The sensitivity of a sensor is defined as the slope of the response curve in its linear range. A chemical sensor is said to be selective if it can discriminate between a selected analyte and other species (said to be “interfering”) within a sample. Increasing sensitivity and selectivity is the main goal driving the use of nanomaterials in chemical sensors. Because of their high surface over volume ratio, nanomaterials are expected to have higher sensitivity. The capability to engineer their composition and crystalline structure at the atomic scale opens up the possibility to design more selective recognition layers.

### 2.2. Carbon Nanotubes (CNTs) Sensors

#### 2.2.1. Carbon Nanotubes Structure

Carbon nanotubes composition, crystalline structure, fabrication and properties have been extensively reviewed over the years [23,24,25,26]. Briefly, carbon nanotubes are more easily described starting from graphene (Figure 1). Graphene consists of a single sheet of carbon atoms arranged in hexagonal cells with an in-plane structure. This form of coordination for carbon atoms is called the sp^2^ hybridization of carbon: the 2s orbital and two of the 2p orbitals (the p_x_ and p_y_) hybridize to form 3 covalent (σ) bonds per carbon atom spread at 120 °C from each other (trigonal structure), while one electron in the p_z_ orbital remains free to contribute to conduction. Carbon nanotubes (CNTs) can then be described as a rolled sheet of graphene with axial symmetry. They are sorted into two main categories, Single-Walled Carbon Nanotubes (SWCNTs) and Multi-Walled Carbon Nanotubes (MWCNTs) depending on the number of graphene layers rolled into a coaxial array (Figure 1). SWCNTs have typical diameters in the range 0.4 to 20 nm and typical lengths from 100 nm to 10 μm depending on the method used to synthesize them. In MWCNTs, each tube is separated from the next by 0.34 to 0.36 nm. MWCNTs may have diameter between 1 nm (double-walled carbon nanotubes (DWCNTs)) to 300 nm (about 100 coaxial tubes) and typical length from 1 μm to 150 μm. In both cases, the length-to-diameter ratio is very high, so they are regarded as 1D nanostructures. SWCNTs can be either semi-conducting or metallic depending on their chirality (the orientation of the lattice with respect to the tube axis) while MWCNT is metallic (except in rare DWCNTs cases).

**Figure 1 sensors-22-00218-f001:**
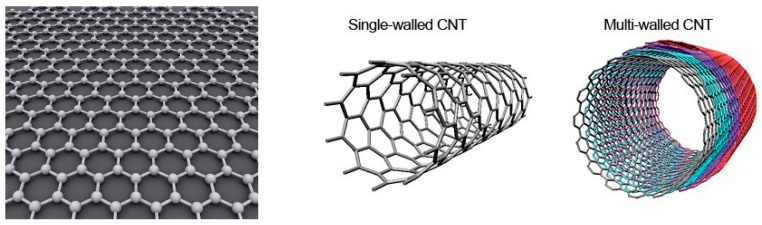
Basic structure of a graphene (**left**) [27], single-walled carbon nanotube (**middle**) and multi-walled carbon nanotube (**right**) [28]. CC BY-SA 3.0.

#### 2.2.2. Functionalization of CNTs

CNTs are very attractive as active materials in chemical sensors as they have high adsorption capability to a wide range of species. However, various results show the limited selectivity of pristine CNT-based chemical sensors (for both gas-phase and liquid-phase sensing): they are often sensitive to different analytes with the same range of magnitude of sensitivity and response time [29,30].

Functionalizing CNT consists in hybridizing them with other molecules either by covalent [31,32] or non-covalent bonds [33,34]. Functionalization is advantageous for selective sensing because the functionalizing molecules can be selected for their affinity to the target analyte. It has become the most popular approach to enhance the selectivity of CNT-based chemical sensors, though some studies report on modulating electrode material instead of functionalizing CNT as a mean to achieve selectivity [35].

One of the main challenges in using functionalization for selective sensing lies in ensuring that the changes occurring at the functionalizing molecules in the presence of the target analyte can be detected through the CNTs in the selected electronic device configuration. At the same time, the functionalization itself should not degrade dramatically the properties of the electronic device itself.

For instance, covalent functionalization is usually expected to allow for stronger charge transfer between CNT and functionalizing molecules, thus providing stronger sensitivity to the target analytes. However, covalent bonds degrade the crystalline structure of the CNTs, thus degrading their conduction properties and subsequently the transduction quality. As a consequence, the density of covalent functionalization that can be achieved in practice remains limited, which in turn may limit the gain in sensitivity and selectivity [36].

By contrast, using non-covalent functionalization, full coverage of the CNT surface may be achieved without degrading the intrinsic electronic properties of the CNTs; however, selecting functionalizing molecules that strongly impact the electronic properties of CNTs is challenging [37]. Usually, molecules that can functionalize CNTs by strong π-stacking are selected among aromatic molecules such as derivatives of benzene, fluorene, carbazole, or porphyrin, or conjugated polymers [38,39].

CNTs often carry some carboxyl groups (-COOH functions) on their sidewalls as a result of the synthesis process or of the post-synthesis purification (see Section 2.3.3). The density of these groups may for instance be evaluated by Raman spectroscopy [40,41], but is not systematically studied in the literature on CNT sensors. Hence CNTs reported on as non-functionalized CNTs may carry COOH groups. The COOH density may also be increased on purpose to enhance sensitivity to certain analytes, for instance by strong oxidative acidic treatments [42]. In this review, CNTs oxidized on purpose are labeled CNT-COOH. Articles reporting on those are classified jointly with the articles on pristine CNTs.

#### 2.2.3. CNT-Based Electronic Devices

Using CNT in electrical sensors requires their integration into electronic devices. This topic has been extensively discussed in the literature (see reference [43] for instance). Briefly, one differentiates between devices based on as-grown CNTs or on prefabricated CNTs. Devices can be either based on a single CNT [44] or a CNT network [45]. In turn, this network may be either random or organized (for instance aligned).

In devices based on as-grown CNTs, CNTs are usually synthesized via chemical vapor deposition (CVD) directly onto pre-patterned electrodes within a temperature range from 550 °C to 1000 °C [46,47]. CVD CNT growth leads to robust electrode/CNT contacts and high CNT crystalline quality while avoiding bundling. However, the high-temperature CVD growth conditions usually prevent the use of flexible substrates. The need for metallic growth catalysts is often incompatible with the architecture of electronic devices (as they require well-defined insulating surfaces). For those reasons, as-grown CNT films are often transferred as a whole from the synthesis substrate onto more appropriate substrates via lift-off [46]. In addition, in-place synthesis does not allow for perfect control of CNT alignment, nor of their diameter, chirality or crystallinity, while these parameters have key impacts on device features. There are several purification and sorting techniques available to tune these parameters for CNTs on solid substrates. The most frequently reported post-growth processes are removal of the metallic CNTs by electrical breakdown [48] (application of a high current to a CNT network while the semiconducting CNTs are polarized in their OFF-state, which burns out metallic CNTs only) or degradation of the CNT crystalline quality by plasma etching [49], irradiation [50], or thermal oxidation [51].

By contrast, in devices relying on pre-fabricated CNTs, CNTs available in powder form are dispersed in a solvent and deposited onto the appropriate substrate via wet process. It is the most frequently reported approach to fabricate CNT-based sensors. It is advantageous because it features little constraints regarding substrates and because it allows the use of a large panel of solution-based CNT pre-treatment protocols, such as purification, acidification, functionalization, sorting by chirality or by diameter [52]. A large variety of techniques is available to deposit CNTs from a dispersion onto a substrate: drop-casting [53], spin coating [54], dip-coating [55], inkjet printing [56], spray-coating [57], aerosol jet printing [58] or vacumm filtering [59]. After substrate deposition of a CNT dispersion, particularly following drop casting, dielectrophoresis may then be used to improve on the deposition quality, notably to control accurately CNT positioning or to achieve CNT alignment [60]. After vaccum filtering, the filter may be removed, forming a freestanding film often called buckypaper [61]. Let us note that, despite the advantages of using prefabricated CNTs compared to in-place growth CNTs, it also has a few drawbacks, such as: CNT placement on the substrate may not be as accurate; low network density is more difficult to achieve; CNT-substrate interaction may be less strong; CNT crystallinity may be degraded during liquid phase processing steps such as high-power sonication.

### 2.3. CNT-Based Electrochemical Sensors

#### 2.3.1. Electrochemical Cells

An electrochemical sensor is a device that detects an electron exchange between sensor and analyte. It is usually composed of two basic components, a chemical recognition layer and a physicochemical transducer, the latter comprising several metal electrodes, the working electrode, the reference electrode and in most cases a counter electrode. Immersed into an electrolyte solution, they make up the electrochemical cell (or voltaic or galvanic cell) [62].

A two-electrode cell consists of only working and reference electrodes. It is used for low current operation (small-sized working electrodes, very low analyte concentrations) because at higher current, the potential of the working electrode becomes unstable. In most applications, a three-electrode cell is used; the reference electrode is maintained at a stable potential, while the current passes through working and counter electrodes. Two types of process may occur in electrochemical cells. In a Faradaic process, charge particles transfer from electrode to electrode through the electrolyte. In non-Faradaic, charge is progressively stored [63].

#### 2.3.2. Electrochemical Transduction

There are various types of electrochemical transducers depending on how the electrochemical cell is operated. The most popular ones are briefly described in the following paragraphs.

In potentiometric sensors, the measured signal is the potential difference between the working electrode and the reference electrode in the absence of current. The working electrode potential depends on the concentration of the target analyte. A reference electrode is needed to provide a defined reference potential. The response of a potentiometric sensor is interpreted using the Nernst equation, which states that the activity of the species of interest is in a logarithmic relationship with the potential difference [64]. This approach works well when the activity of a given species can be approximated to the molar concentration, namely at low concentration.

In voltammetric sensors, the current response is measured as a function of the applied potential. It is directly correlated to the rate of electron transfer occurring via electrochemical reactions [65]. This approach differentiates well species with different redox potential (separated by more than ±0.04–0.05 V). In turn, there are significant interfering effects if two or more species in the sample solution have similar redox potentials. There are different types of voltammetry depending on the way the voltage is applied, notably linear sweep or pulse-wise increase. The latter (usually called differential pulse voltammetry), is reported to be well suited for solid electrodes based on organic compound and more sensitive than the former (usually called cyclic voltammetry).

In electrochemical impedance spectroscopy (EIS), an alternating voltage is applied. The phase shift and amplitude of the current are measured over a range of frequencies. It provides information on the rate of the electrochemical reactions and on the ionic transport in the electrolyte [66].

Stripping voltammetry consists of two steps. First, target chemical species are electrolytically deposited on the surface of one of the electrodes using a constant potential, for instance by reduction of metal ions on the cathode. Second, a voltage scan is applied to the electrode, which progressively strips the target analytes from the electrode depending on their redox potential. At a given voltage, the resulting faradic current is proportional to the concentration of the target chemical ionic species [67]. If the different species are stripped at different voltages, selectivity is possible. The electrode deposition step has a pre-concentration effect on the target analyte, which yields this technique its considerable sensitivity (sub-nanomolar range for metal ions). Figure 2 shows example of measured responses by these three transduction methods.

**Figure 2 sensors-22-00218-f002:**
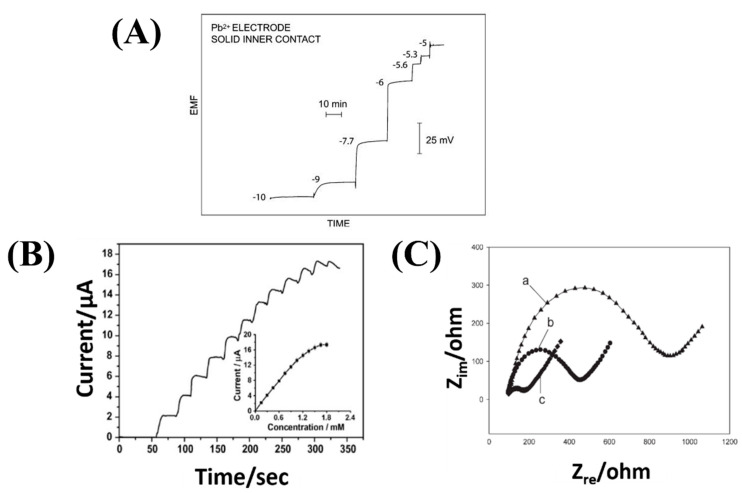
Measured response by three different transduction methods of electrochemical sensors: (**A**) Potentiometry (Potential at zero current—EMF electromotive force—vs. time under increasing volume of analyte—the numbers shown are logarithmic molar sample concentrations. Reproduced from [64] (**B**) Voltammetry (Current versus time under increasing volume of analyte) Reproduced from [68] (**C**) Electrochemical impedance spectroscopy (Nyquist plot: real impedance vs imaginary impedance for different of electrodes; a—Bare glassy carbon electrode (GCE) 1; b—CNTs/poly(1,2-diaminobenzene) prepared by cyclic voltammetry modified GCE; c—CNTs/poly(1,2-diaminobenzene) composite prepared by multipulse potentiostatic method modified GCE). Reproduced from [69].

#### 2.3.3. Use of CNTs in Electrochemical Sensors

Because the performances of electrochemical sensors are driven by the specificities of the electrodes and of the electrolyte/electrode interfaces, improvements in sensor performances can be achieved by tuning either the electrode bulk material or the electrode surfaces, the latter using either dedicated coatings or by surface engineering (for instance, roughness increase). CNTs are used both as coatings and as electrode material to leverage their high specific surface area. It allows for a large dynamic range and for a high loading in electrocatalysts (defined as the catalysts that participate in electrochemical reactions by increasing the rate of chemical reactions without being consumed in the process). Moreover, CNTs display resistance to fouling [70].

The details of the composition and fabrication process of the various CNT-based electrochemical sensors reported in this paper are provided later. As a summary, out of 66 reports, 31 (47%) reports address CNTs coated on glassy carbon electrodes, 10 (15%) are about electrodes directly made out of CNTs or CNT paste, the rest electrodes being made of miscellaneous metallic materials (gold, steel …). The most used method for coating electrodes with CNTs is drop casting (28 references, 42%): CNTs are first purified, then chemically activated (either oxidized or functionalized) and dispersed in a solvent with sonication. The dispersion is then dropped on the electrode surfaces and the solvent is evaporated rapidly [68,71,72,73,74,75,76].

The prevalence of drop-casting methods is due to their simple implementation. They are often used as a stepping stone on the path toward more reproducible, but often less straightforward, fabrication processes. One of the main shortcomings of techniques based on CNT dispersion (drop and spray casting, dip coating, dielectrophoresis, printing …) is that most solvents have low exfoliation efficiency for CNTs and the resulting solutions have low stability due to the rather weak interactions between these solvents and CNTs [77]. As a consequence, CNT-paste-based electrodes are a popular alternative to CNT-coated electrodes (13 references, 20%). The reported binders are often mineral oils, often mixed with graphite powder and/or ionic liquids ([78,79]).

Regarding electrochemical sensing mechanisms, the carbon atoms at the CNT ends have been shown to behave like the edge planes of highly orientated pyrolytic graphite (HOPG) and to feature rapid electron transfer kinetics: they contribute to the Faradaic processes and provide quick response time. By contrast, the carbon atoms of the sidewalls resemble the basal plane of HOPG and show slower electron transfer kinetics than end atoms [80] (though still higher than HOPG due to curvature [81]). In other words, they are much less involved in oxidoreduction reactions with the electrolyte. However, they contribute to non-Faradaic processes driven by adsorption and desorption mechanisms.

The processes enabling removal of the impurities from CNTs that remain from the synthesis process (carbon nanoparticles, nanocrystal metal catalysts, amorphous carbon …) play a strong role in the electrochemical properties of the CNTs. Raw CNTs are usually purified before use by thermal treatment at around 400 °C or by chemical oxidation via acidic treatment. It leads to shortened and partially oxidized CNT. In particular, the resulting CNTs feature functional oxygenated groups at the open ends and an increased defect density along the sidewalls [82]. In addition to CNT curvature, those defects also explain that CNT sidewalls contribute to the Faradaic process in electrochemical sensors. Luo et al. for instance detailed the oxidation–reduction reactions for carboxylic CNT sidewall defects in [83].

### 2.4. CNT-Based Chemistors

#### 2.4.1. Chemistors

Chemi-resistors, or chemistors, are sensors operating by measuring the variation in the electrical resistance or the resistivity of a sensing (also called active) material as a consequence of its interaction with the target analyte. The target analyte has to be in direct contact or close proximity to the active material. The possible interactions are highly diversified: bulk or catalytic reactions, reversible or irreversible, chemi- or physisorption, surface or volume reactions or reactions at grain boundaries [84].

In most chemistors, resistance changes are measured in a two-terminal configuration (Figure 3a,b). A small constant current is applied between two electrodes separated by a short distance (µm to mm) and the resulting voltage is measured. Alternately, four-terminal configurations may also be used to reduce the influence of contact resistance on the sensitivity, especially in the case of high resistance devices (MΩ range and higher). Four parallel electrodes are often used in those cases; the current is applied on the external electrodes and the voltage drop is measured across the two internal electrodes. In the case of arbitrary electrode disposition (for instance, anisotropic surface), the Van der Pauw method can be used to measure the bulk resistivity (ρ) and the Hall coefficient of the surface by using four different contact point [85].

#### 2.4.2. Use of CNT in Chemistors

The use of CNT as active layer and/or electrode material in chemistor is prominent across various sensing applications (gas sensing, biological sensing) [86], as it is the most straightforward device structure available to assess sensitivity of CNTs to chemicals (in terms of design, fabrication, electronics, signal processing …). The high surface area results in high adsorption rates for analytes leading to a rapid response time. Typically, only a fraction of µg or less of CNT material is needed, so the raw material cost is not a limiting factor [87]. Moreover, a small (1 cm^2^) chip-based device can hold hundreds of sensor elements. Such miniaturization leads to a reduction in size and weight of the assembled systems.

Most reports on chemistors in this review (10 out of 13) use SWCNTs rather than MWCNTs. CNT chemistors are mostly often fabricated using electrode materials made of noble metals (platinum and gold), though occasionally (here 4 reports out of 12) the CNTs make up both electrode material and active layer. The electrode metal is usually thermally evaporated on the substrate and patterned with photolithography. After purification (eliminating synthesis residues), sorting (for instance by diameter) and dispersion in a solvent, the CNTs are deposited across the gap, bridging the electrodes or electrode fingers (Figure 3a,b), then the solvent is evaporated. Various methods can be used for this deposition step, either wet-processing techniques (such as drop-casting, inkjet printing, spraying …—11 references) or dry-processing techniques (such as direct (in-place) chemical vapor deposition (CVD) growth or CVD-growth followed by solid-state transfer or by nanoimprint as nano-scale patterning process—2 references). The CNT networks are in most cases random (except in 3 references where there are aligned through dielectrophoresis [88] or threading of CVD-aligned CNTs [89]).

The baseline resistance level of a device and its sensitivity depends on the geometry of the electrodes, on the type and quality of CNTs as well as on their surface density. The latter actually depends on both the CNT concentration in the dispersion and the selected deposition process. The geometry of the electrodes is characterized primarily by their spacing—often called the gap—and the length of the gap. The gap ranges between 1 µm and 100 µm, with the gap length of between 10 µm and several mm. To optimize space occupation, the electrodes are often interdigitated (Figure 3c): instead of a straight gap, the gap is formed by a series of parallel fingers. The effective gap length is thus roughly equal to twice the finger length multiplied by the number of fingers. Finger widths are typically in the 1 to 10 µm range, lengths in the 10 to 100 µm range [90,91].
Figure 3(**a**) Top view and (**b**) cross section of an example of two terminal resistive CNT sensors on ETFE (Ethylene Tetra Fluoro Ethylene). Reproduced from [92]. (**c**) Schematic of interdigitated electrodes.
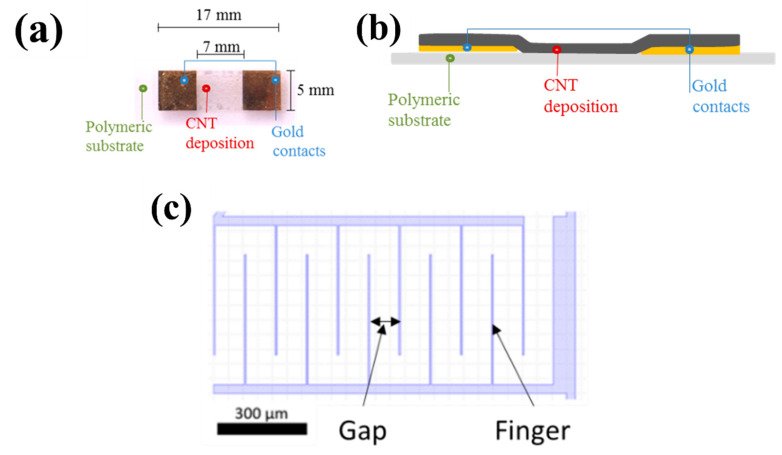



### 2.5. CNT-Based ChemFET

#### 2.5.1. ChemFET

A field-effect transistor (FET) is an electronic device consisting of a semiconducting layer, called channel, located between a source and a drain electrode; the density of electronic carriers flowing in the channel between source and drain electrodes is modulated by the potential of a third electrode, called gate electrode, close to the channel and insulated from it by a dielectric material. A chemical FET—chemFET—is a FET whose conduction characteristics are modulated by the presence and concentration of analytes around the device. The chemically sensitive layer is usually the semiconducting channel, though the electrodes and the dielectric layer have been reported to contribute to sensitivity. The device design allows for the semiconducting layer to be exposed to the target liquid. In an electrolyte-gated chemFET, the target liquid itself is used as both the gate and dielectric layer.

In broad terms, chemFETs are similar to chemistors in the sense that they can be described as chemistors whose baseline resistance is controlled by the gate voltage. Similarly to chemistors, chemical detection is enabled by short-range interactions between the target analyte and the active layer. However, while in chemistors only the resistance (and sometimes the resistivity and contact resistance) of the active layer may be exploited to derive the analyte concentration, there are much more varying parameters in chemFET architectures, which offers a finer understanding of the sensing mechanisms.

In more details, the electrical properties of chemFETs are derived from measuring its drain-source current I_ds_ as a function of gate V_g_ and drain voltage V_d_. From these curves are extracted synthetic parameters such as: (i) in the linear regime of I_ds_ as a function of V_d_, the (gate-voltage controlled chemFET resistance (slope of I_ds_(V_d_)); (ii) the ratio between ON and OFF current levels (values of I_ds_ when the semiconducting channel is respectively in its most conducting—ON—and most insulating—OFF—state); (iii) the transconductance (the maximum value of the first derivative of I_ds_(V_g_), which is related to the mobility of the semiconducting channel); (iv) the threshold voltage (gate voltage value for which the semiconducting channel transitions from insulating to conducting); (v) the hysteresis observed between threshold voltage values or transconductance values during upward and downward sweep of the gate voltage [93,94,95]. Regarding the latter, this hysteresis is attributed in large part to the adsorption of water molecules on the device surface creating charging effect [96]. Hence, it is expected to be a particularly relevant indicator in CNTFET-based water quality sensors but there has very been little study on it so far [97]. Because of this diversity of output parameters, chemFETs are usually considered to be more sensitive and more selective than chemistors. In turn, they usually require significantly more complex fabrication and characterization procedures as well as operating electronics and signal processing.

#### 2.5.2. Use of CNT in ChemFET

CNT-FET is a chemFET with a CNT layer as channel. Because a semiconducting channel is required, only single-walled carbon nanotubes (SWCNTs) can be used [98] (MWCNTs being metallic). In general, the CNT-FET channel may be formed either by a single semi-conducting SWCNT or by a percolating network of SWCNTs with a semi-conducting behavior [99]. While devices based on a single SWCNT have remarkable electrical performances [43,100], devices based on a random-network of SWCNTs are more popular for sensing applications due to their higher effective sensing area, their simpler fabrication procedure as well as their better up-scalability (for mass production), even though their electrical performances are not as good as these of single SWCNT devices. In the field of water quality monitoring, only devices based on networks of CNT have been reported so far. Those networks are in most cases random except in 2 references where CNTs are aligned through dielectrophoresis [60,101].

As SWCNTs may be either semi-conducting or metallic depending on their chiral structure, percolating networks of SWCNTs are not normally semi-conducting because they contain a significant ratio of metallic SWCNTs. To achieve semiconducting SWCNTs networks, a variety of processes is available. For instance, semiconducting SWCNT may be sorted before deposition; electrical breakdown of metallic SWCNTs after their deposition or their in-place growth can also be used; almost systematically, low density networks are used to minimize the chance of forming a metallic path between electrodes [102,103,104].

There are four main types of device architecture for CNT-FET chemical sensors: top gate, bottom gate, liquid gate and hybrid structures; Among 11 reported CNT-FET-based chemical sensors, two are top-gated [105,106], three bottom-gated [107,108,109] and five liquid gated [60,110,111,112,113]. Finally, one is a hybrid dual-gate structure [54] (Figure 4). The original architecture is the bottom gate one, where the gate is embedded below the semiconducting layer with a separating dielectric layer [107]. In the context of water quality monitoring, it has the significant drawback of requiring a high gate voltage (usually several tens of Volts) for good electrical performances, which leads to hydrolysis of water (beyond 1 V). In top gate structures, the gate layer is located on top of the semiconducting channel instead, which makes it more straightforward to fabricate. It requires a lower operating gate voltage, but it is relatively little used for sensing applications as well because the top gate insulates the sensitive channel from the environment. A variation on the top gate structure, the liquid gate structure, consists in applying the gate voltage through the electrolyte surrounding the device [114]. It is particularly interesting for chemical sensing in water because it allows much lower-voltage operation (in the sub-volt range) compared to the usual bottom-gate structure (Figure 4a) [115]. It is also more straightforward to fabricate considering that it requires one less electrode by an embedded gate structure compared to the top gate structure [112]. Hybrid CNT-FET architectures consist in coupling in the same architecture several gating strategies. Notably, Pyo et al. [54] fabricated a double-gated CNT-FET with a separated extended gate. The extended gate concept consists on placing an ion-sensitive membrane on top of the top gate of the double-gated CNT-FET (Figure 4c).

SiO_2_ is the most frequently used dielectric material used for the layer between the semiconducting channel and the substrate or gate electrode (Figure 4b) (6 papers out of 11). However, oxides with a higher dielectric constant such as Al_2_O_3_ or Si_3_N_4_ may also be used in order to have a thinner insulating layer with better homogeneity and durability compared to SiO_2_ [116]. Takeda et al. used a sol-gel layer of 3-aminopropyltriethoxysilane (APS) as insulating layer, with the goal to better immobilize the SWCNTs in water [106]. Indeed, an electrostatic attraction occurs between the negatively-charged carboxyl groups of SWCNTs and the positively-charged APS.

**Figure 4 sensors-22-00218-f004:**
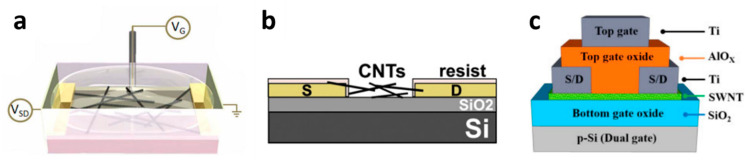
Topology of (**a**) liquid top gate, (**b**) bottom gate and (**c**) extended-dual gate CNT-FET. Reproduced from [54,60,107]. Reprinted with permission. CC BY 3.0.

### 2.6. Sensing Mechanisms in CNT-Based Chemistors and ChemFET

The mechanisms of sensitivity of CNT-based chemistors and chemFET are usually extrapolated from their mechanisms of sensitivity to gas [117], mechanisms that still remain somewhat debated. Overall, the response to analytes is attributed to a change in the conduction properties of either, or all, of the three following components of the devices, as shown in Figure 5 [15]: The conduction along the tube length (“intra-CNT”), the contact points between tubes behaving as tunnel junctions (“inter-CNT”) and the contact points between the tubes and the metal electrodes behaving as Schottky barriers. Sensitivity is attributed either to direct adsorption of the analytes on these sites, or to analytes not adsorbed, but at a distance small enough to these sites to perturb their electrical behavior.

Modulation of the Schottky barrier is caused by a change in the work function of either the electrode metal or the CNTs in presence of the target analytes. The inter-CNT modulation corresponds to a change in the transmission coefficient of the inter-tube tunneling junction, which in turn can be attributed to either a change in the inter-tube distance, or a change in the work functions of the tubes. The intra-CNT conduction modulation is caused by a change either in the density of charge along the sidewalls (resulting in a doping effect) or in the carrier scattering properties of the sidewalls (impacting mobility).

Each of these three modulations may impact the global device response, with specificities depending on the type of transduction (ChemET or chemistor) and on the device morphology (particularly on the network density).

Regarding ChemFETs, Figure 6 provides an insight into the impact of the Schottky-barrier modulation and the intra-CNT modulation (doping or mobility variation) on the typical I-V characteristics of a chemFET [15]. Inter-CNT effects (contact resistance modulation) are usually neglected when analyzing sensing performances of chemFET as their electric performances are mostly driven by intra-CNT effects and Schottky barrier modulation.

By contrast, in chemistors, the inter-CNT modulation is generally accepted to have the strongest impact on the device response, as the global baseline resistance of the network is mostly controlled by inter-CNT contacts. This is confirmed by modeling results for high density networks. In low density networks, modeling suggests that the variations of the electrode-CNT resistance and of the intra-CNT resistance may also contribute to the global relative resistance variation occurring upon exposure to chemicals [118].

## 3. CNT-Based Sensors with Different Analytes in Water

Table 1 summarizes the contents of the 90 reported references on CNT-based sensors considered in this review. The review includes references dealing with all the water-quality relevant analytes discussed until April 2021 except for pesticides, this specific and very large topic having been reviewed very recently elsewhere [119,120]. The most investigated analytes are H^+^ (pH) and lead (with 18% of references each), then cadmium (14%) and nitrite (11%). Altogether, micronutrients and toxic metals cover 37 papers, so 40% of references, a lot of these references covering several analytes.

The large majority of reports addresses MWCNTs (71%), functionalized CNTs (82%) and electrochemical transduction—73%—(14% for chemistors and 12% for ChemFETs).

The following sections provide for each analyte a table with the highlights of the corresponding references. The tables include the following information:Materials: type of CNT (MWCNT or SWCNT), functional probe and type of functionalization (covalent or not),Device strategy: type of transduction (including type of electrochemical measurement and type of FET, gating), CNT deposition process, electrode material and configuration, choice of substratePerformances: limit of detection (LOD) (converted in the most used unit for the target analyte), sensitivity in the measured range of concentration (converted whenever possible in a common unit) and the results of interference study

For each analyte, the best results are figured out for discussions on the choice of functionalization and of the transduction strategies. Whenever possible, the impact of the fabrication strategy is discussed.

### 3.1. pH

pH is the physicochemical quantity defined as pH = −log[H_3_O^+^] [121]. Recommended levels of pH for drinking water are 6.5 to 8.5 according to World Health Organization [122]. Commercial pH sensors for water monitoring applications often cover larger ranges of pH (regularly from 1 to 14) to address abnormal situations regarding drink water (e.g., pollutant ingress) and to be applicable to industrial water processing (chlorination, waste water, aquaculture) as well.

Table 2 shows the 16 reported CNT-based pH sensors: seven are chemistors, six are chemFETs and five are electrochemical sensors (among which two are CNT-FET operated as EC sensors). Only four reports out of 16 use MWCNTs; eight reports address non-functionalized CNTs (including CNT-COOH). This is in contrast with other analytes (as summarized in Table 1), where functionalization is quasi-systematic and the use of chemistors is rare.

By contrast to other analytes also, the use of detection limits (LOD) expressed in M of H^+^ as a mean of comparison between references is challenging because, due to the logarithmic pH scale, these LOD do not translate directly into pH detection limits. Moreover, it is only rarely provided (here seven papers out of 16 only). Because of the variety of reported response types (current, voltage, resistance, conductance, impedance, percentages) the absolute sensitivities (e.g., variation of the response by pH unit) cannot be compared either. As an alternative, we elected here to compare the relative sensitivity at pH 7 (e.g., the variation of the response by pH unit divided by the response at pH 7) which allows for comparison across transduction methods.

Using this indicator, we observed that the 3 non-functionalized and the COOH-based chemistors have considerably better performances than the functionalized ones, the best performance being achieved at 18%/pH unit (63 Ω/pH unit) with MWCNTs sucked by vacuum force on filter paper [123].

For FET as well, the best performance (23%/pH unit) is achieved with spin coated non-functionalized SWCNT in a dual gate chemFET structure [54], the authors showing that double-gated operation performs better than single-gated.

The same performance (23%/pH unit) is achieved with impedance spectroscopy of COOH-functionalized MWCNT spin-coated on Kapton^®^ with gold electrodes [59]. Using Aluminum electrode leads to a significant decrease in performance (14%/pH unit).

These results underline the very good sensitivity to pH of pristine CNTs and CNT-COOH for all three types of transduction methods, while the use of other functional probes degrades performances. This confirms the widespread theory mentioned in Section 2.2.2 that the sensitivity of pristine CNT and CNT-COOH to pH is due to the presence of carboxyl groups on the CNT sidewalls.

It is worth mentioning that the five references on potentiometry yield sensitivities in mV/pH unit very close to Nernst law irrespective of the functionalization (58 mV/pH unit) [54,60,124,125,126].

Finally, one observes that, while there are several reports with the same transduction mode (particularly resistive) and the same functionalization (pristine) [57,58,123,126], it is not possible to draw conclusions regarding to optimal design and fabrication of pH sensors as there is still too much variability in the choice of substrates, electrode materials and deposition method.
sensors-22-00218-t002_Table 2Table 2CNT-based pH sensors in water, sorted by transduction type then by relative sensitivity.Type of CNTFunctional ProbeFunctionalizationAnalyteDetection RangeDetection Limit SensitivityRelative Sensitivity *Transduction MethodCNT Deposition MethodElectrode MaterialContact ConfigurationSubstrateCommentsRef.SWCNTPolyanilineNon covalentpHpH 2.1~12.82.74 nMN/AChemistorDrop-castingTi/AuSi/SiO_2_
[126]SWCNTNafionNon covalentpHpH 1~12N.P.3.5%/pHChemistorScreen printingSWCNTPolymide
[127]MWCNTNi NP *Non covalentpHpH 2~10N.P.5.0%/pHChemistorContinuous pulling of super-aligned, CVD grown MWCNTsMWCNTPDMS
[89]SWCNTPristineNon functionalizedpHpH 1~11<10 pM34 nS/pH3.4%/pH(pH 1~6)163 nS/pH9.3%/pH(pH 7~11)ChemistorSpray-castingCrSi/SiO_2_
[57]SWCNTCOOHCovalentpHpH 5~9N.P.75 Ω/pH11%/pHChemistorDielectrophoresis(aligned CNTs)Cr/AuSi/SiO_2_Response time: 2 s at pH 5, 24 s at pH 9[88]SWCNTPristineNon functionalizedpHpH 4~10N.P.5.2 kΩ/pH14%/pHChemistorAerosol jet printingAgKapton
[58]MWCNTPristineNon functionalizedpHpH 5~9N.P.63 Ω/pH18%/pHChemistorSucked by vacuum forceMWCNTFilter paper
[123]SWCNTMalt extract agarNon covalentpHpH 3~5100 mM N/AFET(hybrid top gate)Dip coatingTi/Au (10/30 nm) contactsSi/SiO_2_(100 nm)Multiplexed detection of Fungus (*A. niger*, *A. versicolor*) and Yeast (*S. cerevisiae*) *[105]SWCNTETH500 *, MDDA-ClNon covalentpHpH 2~7.510 mM71 nA/pH7.5%/pHFET(liquid gate)Spray depositionAqueous electrolyte (gate)Cr/Au (5/50 nm)Polymide (Kapton^®^)Change from p-type to n-type transistor with the membrane layer[112]SWCNTCOOHCovalentpHpH 3~8N.P.17 nA/pH8.2%/pHFET(top gate)N.P.Cr/Au (30/50 nm) source & drain electrodes, Ag/AgCl for reference electrodeGlass/APS(50–200 nm)/SWCNT/APS(500 nm)/TopGateCNT placement controlled by location of APS (modified to immobilize the CNTs)[106]SWCNTPristineNon functionalizedpHpH 3.4~7.810 mM3.9 µA/pH13%/pHFET(bottom gate)Spin coatingCr/Au (5/40 nm)Si/SiO_2_(65 nm)
[107]SWCNTPoly(1-aminoanthracene)Non covalentpHpH 3~111 μMFET19 µS/pH14%/pHpotentiometry55 mV/pHFET, potentiometry(liquid gate)Dielectrophoresis (aligned CNTs)Au contacts, Pt wire (Auxillary), Ag/AgCl electrode (Reference)Si/SiO_2_(300 nm)Multiplexed detection of Ca(II) and Na^+^[60]SWCNTPristineNon functionalizedpHpH 3~101 mM7600 mV/pH23%/pH (Dual-gate mode)59.5 mV/pH (single-gate mode potentiometry)FET(double gate)Spin coating100 nm Ti contacts for source, drain and top gatep-Si (substrate acting as bottom gate)
[54]SWCNTPolyanilineNon covalentpHpH 1~13N.P.56 mV/pHpotentiometrySpray castingPolyvinyl chloride-coated steel wirePVCHighly selective against Li^+^, Na^+^, K^+^[128]MWCNTCOF_THi-TFPB_ *CovalentpHpH 1~12N.P.54 mV/pHDifferential pulse voltammetryDrop castingGlassy carbon electrodeGlassy carbonmultiplexed detection of Ascorbic acid.[125]MWCNTCOOHCovalentpHpH 4~9N.P.17 Ω/pH23%/pH (Au), 16 Ω/pH 14%/pH (Al)Impedance spectroscopyDip coatingAu and Al interdigitated electrodesKapton^®^
[59]* The relative sensitivity is calculated using the formula Relative Sensitivity= (x/x0)∗100 (%), with *x* the absolute sensitivity expressed (depending on the transduction) in units of resistance, voltage or current per pH unit and *x*_0_ the baseline parameter (resistance, voltage or current) at pH 7. The relative sensitivity is not calculated for potentiometry-based transduction as it depends on the choice of reference voltages and the three references can be easily compared by their absolute sensitivity. N.P.: not provided Ni NP: Nickel nanoparticle, PDMS: Polydimethylsiloxane, APS: 3-aminopropyltriethoxysilane, ETH500: tetradodecylammonium tetrakis(4-chlorophenyl)borate, MDDA-Cl: methyltridodecylammonium chloride, *A. niger*: *Aspergillus niger*; *A. versicolor*: *Aspergillus versicolor*, *S. cerevisiae*: *Saccharomyces cerevisiae*, PVC: poly vinyl chloride, COF: Covalent organic framework, Thi: Thionine; TFPB: 1,3,5-tris(p-formylphenyl)benzene.


### 3.2. Micronutrients and Heavy Metals

Micronutrients (Iron, manganese, cobalt, copper, molybdenum, zinc, selenium, cadmium, iodine, boron, fluorine …) are mineral materials that play an important role in metabolic activities and tissue function maintenance in living beings. Subsequently, a suitable intake of micronutrients is necessary, often in trace amounts, and they should not be entirely removed from the water supply. However, they usually become harmful at higher doses and constitute a water quality concern.

Even though various heavy metals, notably cadmium, lead and mercury, are not micronutrients as they are very toxic even in trace amount, a lot of micronutrients actually are heavy metals (notably iron, copper, cobalt and zinc). Hence, micronutrients and heavy metals are often classified jointly in the water quality literature. We proceed in the same manner here.

Metal ions detection in water by CNT-based sensors has been heavily studied since 2005 with 37 papers reported out of 90 papers in total in this review, the large majority of these dealing with electrochemical transduction (33 papers) and functionalized CNTs (31 papers). In the following subsections, we summarize the results on the following ions by order of frequency of occurrence in the literature: Pb(II), Cd(II), Zn(II), Cu(II), Hg(II), As(III), Ni(II) and Co(II). We then discuss the multiplexing performances and the interference studies. The performances of the reported sensors are compared in terms of limit of detection which is the most frequently provided indicator.

#### 3.2.1. Detection of Pb(II)

Table 3 summarizes the different CNT-based Pb(II) sensors used for water quality monitoring. All references but three are based on functionalized MWCNTs sensors using stripping voltammetry, the other ones using stripping voltammetry with pristine MWCNTs [129,130] or potentiometry with functionalized MWCNTs [131,132]. Interestingly for comparison purposes, 6 out of 16 references discuss functionalized MWCNTs drop cast onto glassy carbon electrodes and operated through stripping voltammetry.

The reported ranges of detection cover a large scale, from 0.1 ppt to 100 ppb. By comparison, the maximum acceptable concentration (MAC) of lead worldwide ranges from 10 to 15 ppb [7]. The lowest limits of detection are 0.3 ppb, 0.04 ppb and 0.02 ppt with pristine [129], non-covalently functionalized (Nafion/Bismuth [73]) and covalently functionalized (Dithizone [132]) MWCNTs, respectively.

References [129,130] report the use of pristine CNTs and show that while the limits of detection (0.3 ppb and 1 ppb, respectively) of a pristine CNT electrode are theoretically sufficient for lead detection in the context of water quality monitoring, the ranges of detection are not compatible (lowest limit at respectively 210 ppb and 15 ppb). While ref [129] demonstrates that joint detection of several metal ions is possible (as expected from the principles of stripping voltammetry) both references remark on interferents (dissolved oxygen in [129] and cadmium and copper in [130]), as is expected from pristine nanotubes (non-selective sensing).

Concerning covalent functionalization, references [74,132,133] are based on the same architecture (drop casting of CNT dispersion on glassy carbon and stripping voltammetry), so the major differences in limits of detection can easily be linked to the functionalization strategy. Refs. [74,134] rely on grafting respectively cysteine and thiacalixarene (TCA) on MWCNT by exploiting their sidewall carboxyl groups as reaction sites. The major difference in performances is that TCA is much more favorable for lead complexation than cysteine. Notably, ref. [134] shows by computational method that Pb(II) ions can stably adsorb onto the TCA molecules and that there is a significant electron delocalization between Pb(II) and the sulfur atoms in the TCA molecule. To move beyond the performances of TCA-functionalized CNTs, ref. [132] relies not only on complexation of MWCNTs by dithizone (as thiols have strong interaction with metal ions) but also on the processing of MWCNTs into a bucky-gel, a porous MWCNT-based structure filled with ionic gel (here 1-butyl-3-methylimidazolium hexafluorophosphate). While the functionalization provides reactivity to the target metal ion, the bucky gel is thought to provide strong specific surface area enhancement [135].

In references reporting non-covalent functionalization strategies, bismuth or its derivatives are used in the majority of reports (8 out of 10), the other references citing mercury [131] and antimony oxide [136]. Bismuth is an environmentally friendly material often used as a replacement to mercury in electrochemical applications [137]. Its sensitivity to lead is attributed to its ability to form “fusible” alloys with heavy metals in general. It tends to facilitate their nucleation and subsequent reduction. However, this nucleation process of lead ions around bismuth could in theory limit the reusability of the sensing devices. In practice, Xu et al. (the reference with the best limit of detection among this category) report that the relative standard deviation (RSD) on sensitivity was lower than 5% for Pb(II) detection after 50 repetitive measurements [73].

It is worth noting that the three references about covalent functionalization did not identify any interferent materials, while among seven references reporting on non-covalent functionalization which discussed interferent, only one features no interferent [73], the others mentioned reduced or strong interferents. It suggests that, in the context of lead(II) monitoring, covalent functionalization yields not only better limits of detection but also better selectivity than non-covalent functionalization.

References [138,139] also give the opportunity to discuss the specificity of the stripping voltammetry protocol. Indeed, both references using the exact same materials and process (plating of Bismuth on screen printed CNT electrodes), Injang et al. [138] achieved an improvement of almost one order of magnitude on lead(II) LOD and sensitivity compared to Hwang et al. [139] by complementing anodic stripping voltammetry (ASV) with sequential injection analysis [140]. It is a technique known to enhance sensitivity and selectivity in stripping voltammetry by better controlling reagent and sample volumes. In these two papers, the benefit for this technique occurs only for Pb(II), the two papers reporting similar LOD and sensitivity for Cd(II) and Zn(II) detection (see next sections). This suggests that the optimization of the electrochemical transduction scheme may have a very strong impact on sensitivity and selectivity, possibly stronger than the functionalization choice. However, this impact can rarely be assessed in the literature as it is seldom discussed and a reliable comparison between protocols requires references with very strong similarities in electrode design, fabrication process and material choices.

#### 3.2.2. Detection of Cd(II)

Table 4 presents the 13 CNT sensors reported for Cd(II) detection in water. Similarly to the references reporting the detection of Pb(II), all references but one is based on functionalized MWCNTs sensors using stripping voltammetry, the other ones using stripping voltammetry with pristine MWCNTs [129] or stripping potentiometry with functionalized MWCNTs [131]. All except one paper [141] are common with the reports on Pb(II) sensors in the previous section. Unfortunately, there is no reference about covalently functionalized CNT for cadmium sensing.

The reported detection limits lie between 0.02 ppb and 17 ppb to be compared to the MAC of Cd(II) in water which stands between 3 and 5 ppb. Similarly to the case of lead, ref. [129] reporting the use of pristine CNTs shows an acceptable detection limit (0.23 ppb) but a range of detection not compatible with water quality monitoring (lowest limit at 170 ppb).

For the same reason as for Pb(II) detection, all but five papers out of 13 use functionalization compounds integrating bismuth. The best result in term of LOD (0.02 ppb) is achieved with non-covalent functionalization with Poly(sodium4-styrenesulfonate)-Bismuth (PSS-Bi) [142]. The remarkable LOD is attributed to the wrapping of the PSS polymer around the CNTs, providing a high density of adsorbing sites for metal binding without affecting the electronic properties of the CNTs.

From these references, and those about lead as well, one may note that paste-based approaches do not perform very well overall in terms of limits of detection compared to more traditional processing of MWCNTs.

**Table 3 sensors-22-00218-t003:** CNT-based sensors for detecting (II)Pb(II) ions in water, sorted by type of functionalization then detection limit.

Type of CNT	Functional Probe	Functionalization	Analyte (Add. Analytes)	Detection Limit	Sensitivity (Detection Range)	Transduction Method	Deposition Method	Electrode Material Contact Configuration	Substrate	Interference Study	Ref.
MWCNT	Pristine	Non functionalized	Pb(II) (Cd(II), Zn(II), Cu(II))	0.3 ppb	2.2 nA/ppb (210~830 ppb)	Stripping voltammetry	CNT thread	Metal wire and silver conductive epoxy	Glass capillary	Simultaneous determination of Cd(II), Cu(II), Pb(II) and Zn(II) demonstrated The presence of Dissolved Oxygen changes the calibration law for Cd(II)	[129]
MWCNT	Pristine	Non functionalized	Pb(II)	1.0 ppb	1.5 nA/ppb (15~40 ppb) 3.5 nA/ppb (40~70 ppb)	Stripping voltammetry	Inkjet printing	Inkjet-printed silver ink	PEN *	Effects of copper and cadmium are reported.	[130]
MWCNT	Ionic liquid—dithizone based bucky-gel	Covalent	Pb(II)	0.02 ppt	0.024 µA/ppb (0.1ppt~210 ppb)	Stripping voltammetry	Drop-casting	Glassy carbon electrode	Glassy carbon	No interference of Cd(II) and Cu(II) ions with the detection of Pb(II) ion.	[132]
MWCNT	Thiacalixarene	Covalent	Pb(II)	8 ppt	3.8 µA/ppb (0.04–2.07 ppb)	Differential pulse anodic stripping voltammetry	Drop casting	Glassy carbon electrode	Glassy carbon	Detection of Pb(II) was clearly not affected by Zn(II), Cd(II), Ni(II) (100-fold excess)	[134]
MWCNT	Cysteine	Covalent	Pb(II) (Cu(II))	1 ppb	0.23 * µA/ppb (25~750 ppb)	Differential pulse anodic stripping voltammetry	Drop casting	Glassy carbon electrode	Glassy carbon.	40-fold Cl^−^, 30-fold SO_4_^2−^ and four fold CO_3_^2−^ did not have any significant effect on the stripping peak current of Pb(II) and Cu(II)	[74]
MWCNT	Poly(o-toluidine) Ce(III)tungstate	Covalent	Pb(II)	210 ppb	27 mV/decade (0.1 ppt–100 ppb)	Potentiometry	Liquid mixing and membrane formation through drying	Calomel electrode	Glass tube (araldite)	Strong selectivity (from 50 to 500 times) against Zn(II), Sr(II), Hg(II), Ca(II), Pd(II), Cu(II), Mg(II)	[133]
MWCNT	Nafion/Bismuth	Non covalent	Pb(II), (Cd(II))	25 ppt	0.22 µA/ppb (0.05 to 5 ppb) 0.27 µA/ppb (5~100 ppb)	Stripping voltammetry	Drop casting	Glassy carbon electrode	Glassy carbon	500-fold of SCN^−^, Cl^−^, F^−^, PO_4_^3−^, SO_4_^2−^, NO_3_^−^ and various cations such as Na^+^, Ca(II), Mg(II), Al(III), K^+^, Zn(II), Co(II) and Ni(II) had no influences on the signals of Pb(II) and Cd(II).	[73]
MWCNT	PSS-Bi *	Non covalent	Pb(II) (Cd(II))	0.04 ppb	0.079 µA/ppb (0.5~90 ppb)	Stripping voltammetry	Drop casting	Glassy carbon electrode	Glassy carbon	20-fold amounts of Zn(II), 5-fold amounts of Sn(II) and 1-fold amounts of Cu(II) have influence on the determination of Cd(II) and Pb(II) with deviation of 10%.	[142]
MWCNT	Bismuth	Non covalent	Pb(II) (Cd(II))	~0.04 ppb	N/A	Stripping voltammetry	Plasma-enhanced CVD (vertically aligned MWCNTs in epoxy matrix)	Cr	Silicon	N.P. *	[143]
MWCNT	Fe_3_O_4_-LSG-CS-Bi *	Non covalent	Pb(II) (Cd(II))	0.07 ppb	0.21 µA/ppb (1~20 ppb) 0.24 µA/ppb (20~200 ppb)	Stripping voltammetry	Drop casting	Glassy carbon electrode	Glassy carbon	Slight changes in peak currents of Pb(II) and Cd(II) were observed in presence of interfering ions Na^+^, Cl^−^, SO_4_^2−^, PO_4_^3−^, Fe(II), Fe(III), Zn(II), As(III). Significant increase in response signals of Hg(II) was probably due to the formation of amalgam Dramatically decreased response signals of Cu(II) were ascribed to the formation of Pb-Cu inter-metallic compounds.	[144]
MWCNT	PPy-Bi NPs *	Non covalent	Pb(II) (Cd(II))	0.1 ppb	1.1 µA/ppb (0.11~120 ppb)	Stripping voltammetry	Paste mixture with MWCNT, paraffin oil and graphite powder	Stainless steel rod	Teflon (PTFE *) tube	Good selectivity towards Fe(II), Al(III), Zn(II), Mg(II), SO_4_^2^^−^, CO_3_^2^^−^, Ca(II), K^+^, Na^+^. The absolute relative change of signal varied from 0.40 to 4.88%). High interference from Cu(II) (1-fold mass ratio was found as the tolerance ratios for the detection of Pb and Cd ions)	[145]
MWCNT	rGO-Bi *	Non covalent	Pb(II) (Cd(II))	0.2 ppb	930 nA/ppb cm^2^ (20~200 ppb)	Stripping voltammetry	Spray coating	Cr(30 nm)/Au(200 nm)	Polymide (VTEC 1388)	100-fold K^+^, Na^+^, Ca(II), Cl^−^, NO^3−^ and a 30-fold Fe(III) increase had no significant effect on the signals of Cd and Pb ions. Cu ions were found to reduce the response of target metal ions due to the competition between electroplating Bi and Cu on the electrode surface (close reduction potential of Cu and Bi.)	[146]
MWCNT	Bismuth	Non covalent	Pb(II) (Cd(II),Zn(II))	0.2 ppb	0.39 µA/ppb (2~18 ppb) 0.67 µA/ppb (20~100 ppb)	Stripping voltammetry	Screen printing	Screen printed MWCNT based electrode	Ceramic substrates	N.P.	[138]
MWCNT	Bismuth	Non covalent	Pb(II) (Cd(II),Zn(II))	1.3 ppb	1.2 µA/ppb (2~100 ppb)	Stripping voltammetry	Screen printing	Screen printed MWCNT based electrode	Alumina plates	The addition of copper ions strongly influenced the stripping responses. Decrease of lead and cadmium pics by 65.5%.	[139]
MWCNT	Pristine	Non covalent	Pb(II) (Cd(II), Zn(II))	6.6 ppb	0.47 * s/V/ppb (58~650 ppb)	Stripping potentiometry	Paste mixture of MWCNT and mineral oil	MWCNT paste electrode	Glass tube	Al(III), Mg(II), Fe(III), Ni(II), Co(II), Cr(III), Cu(II) and Sb(III) were investigated in the ratio analyte: Interferent 1:1 and 1:10. The interference was observed for the ratios analyte: interferent 1:1 and 1:10 for Co(II), 1:10 for Cr(III) and Cu(II).	[131]
MWCNT	Sb_2_O_3_ *	Non covalent	Pb(II) (Cd(II))	24 ppb	2.7 µA/ppb (5–35 ppb)	Stripping voltammetry	Paste mixture of MWCNT, silicon oil, Sb_2_O_3_ powder and ionic liquid	Copper wire	PTFE tube	N.P.	[136]

* N.P.: Not provided. PSS-Bi: Poly(sodium 4-styrenesulfonate)-Bismuth. rGO-Bi: Reduced graphene oxide-Bismuth. PPy-Bi NPs: Polypurrole-Bismuth NanoParticles. PTFE: poly tetra fluoro ethylene. LSG-Cs-Bi: laser-scribed graphene-chitosan-Bismuth. Sb_2_O_3_: antimony oxide. PEN: polyethylene naphthalate. 1 μM Pb(II) = 210 ppb Pb(II).

**Table 4 sensors-22-00218-t004:** CNT-based sensors for detecting Cd(II) ions in water, sorted by type of functionalization then by detection limit.

Type of CNT	Functional Probe	Functionalization	Analyte (Add. Analytes)	Detection Limit	Sensitivity (Linear Range)	Transduction Method	Deposition Method	Electrode Material Contact Configuration	Substrate	Interference Study	Ref.
MWCNT	Pristine	Non functionalized	Cd(II) (Pb(II) Zn(II), Cu(II))	0.23 ppb	3.9 nA/ppb (170~500 ppb)	Stripping voltammetry	CNT thread	Metal wire and silver conductive epoxy	Glass capillary	Simultaneous determination of Cd(II), Cu(II), Pb(II) and Zn(II) demonstrated The presence of Dissolved Oxygen changes the calibration law for Cd(II)	[129]
MWCNT	PSS-Bi *	Non covalent	Cd(II) (Pb(II))	0.02 ppb	0.23 µA/ppb (0.5~50 ppb)	Stripping voltammetry	Drop casting	Glassy carbon electrode	Glassy carbon	20-fold amounts of Zn(II), 5-fold amounts of Sn(II) and 1-fold amounts of Cu(II) have influence on the determination of Cd(II) and Pb(II) with deviation of 10%.	[142]
MWCNT	Nafion/Bismuth	Non covalent	Cd(II) (Pb(II))	0.04 ppb	0.18 µA/ppb (0.08~5 ppb) 0.16 µA/ppb (5~100 ppb)	Stripping voltammetry	Drop casting	Glassy carbon electrode	Glassy carbon	500-fold of SCN^−^, Cl^−^, F^−^, PO_4_^3−^, SO_4_^2−^, NO_3_^−^ and various cations such as Na^+^, Ca(II), Mg(II), Al(III), K^+^, Zn(II), Co(II) and Ni(II) had no influences on the signals of Pb(II) and Cd(II).	[73]
MWCNT	Bismuth	Non covalent	Cd(II) (Pb(II))	0.04 ppb	0.037 µA/ppb (0.5~8 ppb)	Stripping voltammetry	Plasma-enhanced CVD (vertically aligned MWCNTs in epoxy matrix)	Cr	Silicon	N.P.	[143]
MWCNT	Fe_3_O_4_-LSG-CS-Bi *	Non covalent	Cd(II) (Pb(II))	0.1 ppb	0.097 µA/ppb (1~20 ppb) 0.32 µA/ppb (20~200 ppb)	Stripping voltammetry	Drop casting	Glassy carbon electrode	Glassy carbon	Slight changes in peak currents of Pb(II) and Cd(II) were observed in presence of interfering ions Na^+^, Cl^−^, SO_4_^2−^, PO_4_^3−^, Fe(II), Fe(III), Zn(II), As(III). Significant increase in response signals of Hg(II) was probably due to the formation of amalgam Dramatically decreased response signals of Cu(II) was ascribed to the formation of Pb-Cu inter-metallic compounds.	[144]
MWCNT	PPy-Bi *	Non covalent	Cd(II) (Pb(II))	0.16 ppb	0.47 µA/ppb (0.16~120 ppb)	Stripping voltammetry	Paste mixture with MWCNT, paraffin oil and graphite powder	Stainless steel rod	Teflon (PTFE) tube	Good selectivity towards Fe(II), Al(III), Zn(II), Mg(II), SO_4_^2−^, CO_3_^2−^, Ca(II), K^+^, Na^+^. The absolute relative change of signal varied from 0.40 to 4.88%). High interference from Cu(II) (1-fold mass ratio was found as the tolerance ratios for the detection of Pb and Cd ions)	[145]
MWCNT	Poly(1,2-diaminobenzene)	Non covalent	Cd(II), (Cu(II))	0.25 ppb	0.14 µA/ppb (5~100 ppb)	Stripping voltammetry	Multipulse potentiostatic method	Glassy carbon electrode	Glassy carbon	N.P.	[69]
MWCNT	rGO-Bi *	Non covalent	Cd(II) (Pb(II))	0.6 ppb	26 nA/ppb cm^2^ (20~200 ppb)	Stripping voltammetry	Spray coating	Cr(30 nm)/Au(200 nm)	Polymide (VTEC 1388)	100-fold K^+^, Na^+^, Ca(II), Cl^−^, NO^3-^ and a 30-fold Fe(III) increase had no significant effect on the signals of Cd and Pb ions. Cu ions were found to reduce the response of target metal ions due to the competition between electroplating Bi and Cu on the electrode surface (close reduction potential of Cu and Bi.)	[146]
MWCNT	Bismuth	Non covalent	Cd(II) (Pb(II),Zn(II))	0.7 ppb	0.22 µA/ppb (2~18 ppb) 1.5 µA/ppb (20~100 ppb)	Stripping voltammetry	Screen printing	Screen printed MWCNT based electrode	Alumina plates	The addition of copper ions strongly influenced the stripping responses. Decrease of lead and cadmium pics by 65.5%.	[139]
MWCNT	Bismuth	Non Covalent	Cd(II) (Pb(II),Zn(II))	0.8 ppb	0.59 µA/ppb (2~18 ppb) 0.80 µA/ppb (20~100 ppb)	Stripping voltammetry	Screen printing	Screen printed MWCNT based electrode	Ceramic substrates	N.P.	[138]
MWCNT	Fe_3_O_4_/ eggshell	Non covalent	Cd(II)	2.4 ppb	19 µA/ppb (0.5~210 ppb)	Stripping voltammetry	Paste mixture of MWCNT, graphite powder, paraffin oil and Fe3O4-eggshell	Copper wire	Glass tube	500-fold amounts of the following ions: Na^+^, Ca(II), Mg(II), Fe(III), Mn(II), Cr(III), Ba(II), Co(II), Hg(II), K^+^, NH^4+^, NO_3_^−^, SO_4_^2−^, PO_4_^3−^ made no alteration of the peak currents of Cd(II). 100-fold amounts of Sn(II) and Cu(II) with deviation of 9%, 50 fold amounts of Ni(II) and Zn(II) with deviations of 8% and 6% respectively had influence on the determination of Cd(II).	[141]
MWCNT	Pristine	Non covalent	Cd(II) (Pb(II),Zn(II))	8.4 ppb	0.36 * s/V/ppb (58~646 ppb)	Stripping potentiometry	Paste mixture of MWCNT and mineral oil	MWCNT paste electrode	Glass tube	Al (III), Mg (II), Fe (III), Ni (II), Co (II), Cr (III), Cu (II) and Sb (III) were investigated in the ratio analyte: Interferent 1:1 and 1:10. the interference was observed for the ratios analyte: interferent 1:1 and 1:10 for Co (II), 1:10 for Cr (III) and Cu (II).	[131]
MWCNT	Sb_2_O_3_	Non covalent	Cd(II) (Pb(II))	17 ppb	1.9 µA/ppb (80~150 ppb)	Stripping voltammetry	Paste mixture of MWCNT, silicon oil, Sb_2_O_3_ powder and ionic liquid	Copper wire	PTFE tube	N.P.	[136]

* PSS-Bi: Poly(sodium 4-styrenesulfonate)-Bismuth. rGO-Bi: Reduced graphene oxide-Bismuth. PPy-BiNPs: Polypurrole-Bismuth. LSG-Cs-Bi: laser scribed graphene-chitosan-Bismuth. Sb_2_O_3_: antimony oxide. 1 μM Cd(II) = 112 ppb Cd(II).

#### 3.2.3. Detection of Zn(II)

The maximum acceptable concentration of Zn(II) in water is 5 ppm. Detection of the Zn(II) ion in water using CNT sensors has only been investigated with electrochemical, MWCNT-based sensors (Table 5). The four references have been listed in the previous tables as they address also lead and cadmium. Unlike results on lead and cadmium where functionalized CNTs have much better performances than non-functionalized CNTs, one observes here that one of the two pristine MWCNTs references [129] has a remarkably better limit of detection (0.09 ppb—two orders of magnitude lower) than the two references with MWCNTs non covalently functionalized with bismuth. By contrast, the other reference with pristine MWCNT has worse detection limit than the Bi-functionalized devices. It points out once again the strong sensitivity of water quality sensor performances to CNT processing (CNT thread versus CNT paste) and to transduction mode (stripping voltammetry versus potentiometry). 

#### 3.2.4. Detection of Hg(II)

Table 6 presents CNT-based Hg(II) sensors that were reported to date. The literature is more varied in terms of transduction methods than for Pb(II), Cd(II) and Zn(II): one chemFET and one chemistor-based approaches (with pristine SWCNT) are proposed in addition to three references reporting stripping voltammetry and two others dealing with potentiometry (the last five reports being with functionalized MWCNT).

Only one paper reports the use of a chemistor. It uses pristine SWCNTs and it addresses the ppm range (LOD 0.6 ppm; range 1 to 30 ppm) which is not truly relevant for drink water monitoring as the maximum acceptable concentration of mercury allowed in water is 1 ppb [147]. By comparison, the pristine SWCNT-based chemFET structure provides widely improved performances (LOD of 2 ppb, range 0.2–200 ppm—still a bit high for drink water application). While an improvement on performances is expected when switching from chemistor to CNTFET, such a magnitude (two orders of magnitude of improvement) is usually not. It may be linked to the use of an octadecyl-trichlorosilane (OTS) self-assembled monolayer (SAM) in [110] to favor the adsorption of SWCNT on the SiO_2_ substrate. This type of SAM is reported to improve FET electronic transport performances [148]. Interestingly, both references report very good selectivities despite the absence of functionalization, which suggests a strong natural affinity of pristine CNT to mercury. This is actually confirmed by studies that report water purification using CNTs and show remarkable adsorption capability of mercury by non-functionalized CNTs (containing CNT-COOH or -OH) [149].

It was observed that the LOD drops significantly and the range of detection shifts towards lower (and more relevant) detection limits when functionalization and electrochemical transduction are used. Covalent functionalization with thiophenol brings the LOD down to 0.6 ppb (range from 1 to 18 ppb) [150]. The lowest reported LOD is 2 ppt (range 2 ppt to 1000 ppm) [151]. It was reached (with differential pulse voltammetry) by functionalizing a 3D structure made of MWCNTs randomly arranged around graphene oxide sheets with bismuth-doped polyaniline chains (PANI). Once again, bismuth is used successfully for its ability to interact with heavy metals, while the 3D scaffold is thought to enhance the specific surface area.

**Table 5 sensors-22-00218-t005:** CNT-based sensors for detecting Zn(II) ions in water, sorted by detection limit.

Type of CNT	Functional Probe	Functionalization	Analyte (Add. Analytes)	Detection Limit	Sensitivity (Linear Range)	Transduction Method	Deposition Method	Electrode Material Contact Configuration	Substrate	Interference Study	Ref.
MWCNT	Pristine	Non functionalized	Zn(II) (Cd(II), Pb(II), Cu(II))	0.08 ppb	3.4 pA/ppb (200~590 ppb)	Stripping voltammetry	CNT thread	Metal wire and silver conductive epoxy	Glass capillary	Simultaneous determination of Cd(II), Cu(II), Pb(II) and Zn(II) demonstrated The presence of Dissolved Oxygen changes the calibration law for Cd(II)	[129]
MWCNT	Bismuth	Non covalent	Zn(II) (Pb(II), Cd(II))	11 ppb	0.18 µA/ppb (12~18 ppb) 0.24 µA/ppb (20~100 ppb)	Stripping voltammetry	Screen printing	Screen printed MWCNT based electrode	Ceramic substrates	N.P.	[138]
MWCNT	Bismuth	Non covalent	Zn(II) (Pb(II), Cd(II))	12 ppb	0.38 µA/ppb (20~100 ppb)	Stripping voltammetry	Screen printing	Screen printed MWCNT based electrode	Alumina plates	The addition of copper ions strongly influenced the stripping responses. Decrease of lead and cadmium pics by 65.5%.	[139]
MWCNT	Pristine	Non covalent	Zn(II) (Pb(II), Cd(II))	28 ppb	0.11 * s/V/ppb (58~646 ppb)	Stripping potentiometry	Paste mixture of MWCNT and mineral oil	MWCNT paste electrode	Glass tube	Al(III), Mg(II), Fe(III), Ni(II), Co(II), Cr(III), Cu(II) and Sb(III) were investigated in the ratio analyte: Interferent 1:1 and 1:10. the interference was observed for the ratios analyte: interferent 1:1 and 1:10 for Co(II), 1:10 for Cr(III) and Cu(II).	[131]

* 1 μM Zn(II) = 65 ppb Zn(II).

**Table 6 sensors-22-00218-t006:** CNT-based sensors for detecting Hg(II) ions in water, sorted by type of functionalization, then detection limit.

Type of CNT	Functional Probe	Functionalization	Analyte (Add. Analytes)	Detection Limit	Sensitivity (Linear Range)	Transduction Method	Deposition Method	Electrode Material Contact Configuration	Substrate	Interference Study	Ref.
SWCNT	Pristine	Non functionalized	Hg(II)	0.6 ppm	12 mV/ppm (1~30 ppm)	Chemistor	CVD	SWCNT	Glass	1000 fold excess of Fe(II), Fe(III), Ni(II), Cu(II),Zn(II), Cr(III) and 500 folds of As(III), Sb(III), Se(IV) and Pb(II) had no interfering effect in the analysis of mercury solution.	[147]
SWCNT	Pristine	Non functionalized	Hg(II)	2 ppb	0.22/decade 0.2 ppb~201 ppm	FET (Liquid gate)	Dip coating with selective CNT placement	Pd/Au (10/30 nm)	Glass	Good selectivity towards interferent ions (only Hg(II) causes conductance increase.)	[110]
SWCNT	Thiophenol	Covalent	Hg(II)	0.6 ppb	0.14 µA/ppb (1~18 ppb)	Stripping voltammetry	Dip coating	Au	Au	The presence of 100-fold concentration of Cr(II), Mn(II), Co(II), Ni(II), Zn(II), 50-fold concentration of Fe(II),and 20-fold Cu(II), have no influence on the signals of 50 nM Hg(II) with deviation below 5%.	[150]
MWCNT	PANi-Bi NPs@GO *	Non covalent	Hg(II) (Cu(II))	2 ppt	1.3 µA/ppb (2 ppt~1000 ppm)	Differential pulse voltammetry	Screen printing	(commercial) Carbon ink	PET	Not provided	[151]
MWCNT	Au NPs	Non covalent	Hg(II)	0.06 ppb	0.59 µA/ppb (0.1~1 ppb) 0.045 µA/ppb (1~250 ppb)	Stripping voltammetry	Drop casting	Glassy carbon electrode	Glassy carbon	Not provided	[72]
MWCNT	* ENTZ	Non covalent	Hg(II)	0.5 ppb	29.3 mV/decade (1 ppb~20 ppm)	Potentiometry	Paste mixture of MWCNT, graphite powder, ENTZ ionophore and ionic liquid	Copper wire	Polypropylene syringe	The interfering ions (Ag^+^, Zn(II), Pb(II), Ni(II), Cd(II)and Cu(II)) do not have any effect on the response of proposed electrodes to Hg(II)	[79]
MWCNT	Thiol-functionalized chitosan	Non covalent	Hg(II)	0.6 ppb	1060 µA/ppb (2~28 ppb)	Stripping voltammetry	Drop casting	Glassy carbon electrode	Glassy carbon	100-fold Cd(II)^,^ 100-fold Pb(II), 50-fold Zn(II), 25-fold Cu(II), 10-fold Ag(II), 10-fold Fe(II) and 10-fold Mn(II) caused within ±5% changes of voltammetric signals for Hg(II).	[152]
MWCNT	Triazene (BEPT)	Non covalent	Hg(II)	0.62 ppb	29 mV/decade (0.8 ppb~440 ppm)	Potentiometry	Paste mixture of MWCNT, graphite powder, Triazene (BEPT) ionophore and paraffin oil	Copper wire	Polyethylene tube	The proposed electrode has a high performance to selective potentiometric assay of Hg(II) in aqueous samples containing some interfering ions (Cu(II), Ag(II), Cd(II), Co(II), Al(III), Pb(II), K^+^.	[153]

* ENTZ: 1-(2-ethoxyphenyl)-3-(3-nitrophenyl)triazene. PANI: polyaniline. GO: graphene oxide. NP: nanoparticles. 1 μM Hg(II) = 200 ppb Hg(II).

#### 3.2.5. Detection of As(III)

Table 7 presents the five stripping voltammetry sensors reported for the detection of As(III) in water using functionalized MWCNTs. No report was found on As(V) detection even though it is a relevant species for water quality monitoring. The maximum acceptable concentration of arsenic in drink water is 10 ppb.

COOH-functionalized CNTs [154] are sensitive to As(III) in the proper range of concentration (0.3 to 50 ppb), though the LOD is not provided. It suggests that pristine CNT (which naturally carry COOH groups on their sidewalls) are sensitive to As(III) as well, though the reference reports on interference with antimony.

Four references describe non-covalent functionalization, all but one [155] use metal nanoparticles (NP) as functional probes, as they have been reported to provide good performances for Arsenic detection in water [156]. More specifically, the best LOD is achieved with Au-NP with a value of 0.1 ppb [157]. The enhanced performances of gold NPs compared to others metal NPs for heavy metal detection are usually attributed to their high electrical conductivity, high surface area and catalytic activity. A comparable LOD (0.13 ppb) is achieved with a Leucine/Nafion functionalization [155]. The Leucine peptide is known for its capability to coordinate metal ions through hydrogen bonds occurring between its -NH(III) and -COOH groups.

**Table 7 sensors-22-00218-t007:** CNT-based sensors for detecting As(III) ions in water, sorted by detection limit.

Type of CNT	Functional Probe	Functionalization	Analyte (Add. Analytes)	Detection Limit	Sensitivity (Linear Range)	Transduction Method	Deposition Method	Electrode Material Contact Configuration	Substrate	Interference Study	Ref.
MWCNT	COOH	Covalent	As(III)	N.A	0.24 µA/ppb (0.3~50 ppb)	Stripping voltammetry	Dip coating	Au	Au electrode	Interference was significant when the Sb/As ratio is higher than 1.	[154]
MWCNT	Au-NP *	Non covalent	As(III)	0.1 ppb	26 µA/ppb (75 ppt–5.3 ppm)	Stripping voltammetry	Drop casting	Glassy carbon	Glassy carbon	Not provided	[157]
MWCNT	Leucine/Nafion	Non covalent	As(III)	0.13 ppb	0.27 µA/ppb (0.37~150 ppb)	Stripping voltammetry	Drop casting	Pt	Pt electrode	Zn(II) and Fe(II) could be tolerated up to at least 0.05 mM whereas commonly encountered matrix components such as Cd(II), Co(II), Mg(II), Ni(II) and Cu^+^ did not show high percentage of interference.	[155]
MWCNT	Pt-Fe NP	Non covalent	As(III)	0.75 ppb	64 nA/ppb (0.75~22 ppb)	Stripping voltammetry	Drop casting	Glassy carbon	Glassy carbon	No interference from copper ion	[158]
MWCNT	Au-NPs	Non covalent	As(III)	0.75 ppb	2.6 Q/mL/ppb * (0.75~750 ppb)	Stripping voltammetry	Vacuum filtration	MWCNT membrane	PTFE membrane	The presence of copper at 10 µM strongly affects the analytical response of As(III); The presence of Pb(II) caused a minor broadening of the peak of As(III) resulting in a slight reduction of the peak current;	[159]

* NP: nanoparticles. Q/mL: charge at the peak current by mL of solution passing through (the conversion to A/ppb was not possible with provided information). 1 μM As(III) = 75 ppb As(III).

#### 3.2.6. Detection of Cu(II)

Table 8 shows the results of the nine reported papers for Cu(II) ions detection. All but one [129] (mentioned before regarding Zn(II), Cd(II) and Pb(II) detection) address functionalized CNTs-based sensors. All but one paper address stripping voltammetry with MWCNTs. The remaining one [113] achieves with a peptide-functionalized-SWCNT-FET structure a value of 3 ppt for the LOD over the range 0.6–600 ppt, which is the best one ever reported. The authors tested different combinations of peptides (of which there are in theory unlimited numbers) to identify the one with optimal sensitivity. The approach was also tested successfully for Ni(II) detection (see next section).

It should be noted that the MAC of Cu(II) in drink water is 1 ppm, so the other references targeting the ppb to ppm range with LOD in the ppb range are more relevant to drink water applications. Comparable LOD of 0.01 ppb, 0.02 ppb and 0.03 ppb are achieved respectively with Schiff base [160], pristine [129] and 2-amino-4-thiazoleacetic acid [161] functionalization and stripping voltammetry. While it suggests again a strong natural affinity of CNTs to Cu(II), both functional probes are found interesting as they carry amine groups which are well known to easily complex copper ions [162]. 

**Table 8 sensors-22-00218-t008:** CNT-based sensors for detecting Cu(II) ions in water, sorted by detection limit.

Type of CNT	Functional Probe	Functionalization	Analyte (Add. Analytes)	Detection Limit	Sensitivity	Transduction Method	Deposition Method	Electrode Material Contact Configuration	Substrate	Interference Study	Ref.
SWCNT	PANI-GGHH *	Non covalent	Cu(II)	3 ppt	N/A (3~29 ppt)	FET (liquid gate)	CVD	300 nm Au	Si/SiO_2_ (120nm)	His_6_ shows higher chelation power for Ni(II) than to Cu(II).	[113]
MWCNT	C_24_H_30_N_6_ Schiff base	Non covalent	Cu(II)	10 ppt	N/A (0.09~340 ppb)	Stripping voltammetry	Paste of MWCNT, Schiff base and mineral oil	Copper wire	Filter membrane	Not provided	[160]
MWCNT	Pristine	Non functionalized	Cu(II) (Cd(II), Zn(II), Pb(II))	17 ppt	9.4 pA/ppb (32~220 ppb)	Stripping voltammetry	CNT thread aspirated into a glass capillary	Metal wire and silver conductive epoxy	Glass capillary	Simultaneous determination of Cd(II), Cu(II), Pb(II) and Zn(II) demonstrated The presence of Dissolved Oxygen changes the calibration law for Cd(II)	[129]
MWCNT	2-amino-4-thiazoleacetic acid	Non covalent	Cu(II)	30 ppt	0.02 µA/ppb * (44 ppb~3.2 ppm)	Stripping voltammetry	Drop casting	Glassy carbon electrode	Glassy carbon	At a concentration ratio below 10, the presence of Zn(II), Mn(II), Ni(II), Co(II) has led to lower than 6% decreasing of DPSV currents of Cu(II)^.^	[161]
MWCNT	PANi-Bi NPs@GO *	Non covalent	Cu(II) (Hg(II))	32 ppt	0.23 uA/ppb (32 ppt~320 ppm)	Differential pulse voltammetry	Screen printing	(commercial) Carbon ink	PET	Not provided	[151]
MWCNT	N-doped carbon spheres	Non covalent	Cu(II)	92 ppt	0.28 µA/ppb (0.5~200 ppb)	Stripping voltammetry	Drop casting	Glassy carbon electrode	Glassy carbon	EDTA can seriously affect the stripping peak current of Cu(II) with a decrease of 79%.	[163]
MWCNT	Poly(1,2-diaminobenzene)	Non covalent	Cu(II) (Cd(II))	0.33 ppb	0.11 µA/ppb (5~100 ppb)	Stripping voltammetry	Multipulse potentiostatic method	Glassy carbon electrode	Glassy carbon	Not provided	[69]
MWCNT	SSA/MoS_2_*	Non covalent	Cu(II)	3.6 ppb	0.13 µA/ppb (6.4~−700 ppb)	Stripping voltammetry	Drop casting	Glassy carbon electrode	Glassy carbon	10-fold concentration of the metal ions (K^+^, Ca(II), Na^+^, Mg(II), Zn(II), Pb(II), Cd(II), Fe(III), Mn(II), Co(II), Cr(III), Cr^6+^, Ni(II) and Hg(II), has not any obvious effect on the Cu(II) peak current.	[164]
MWCNT	Cysteine	Covalent	Cu(II) (Pb(II))	15 ppb	0.13 * µA/ppb (250~1500 ppb)	Differential pulse anodic stripping voltammetry	Drop casting	Glassy carbon electrode	Glassy carbon	40-fold Cl^−^, 30-fold SO_4_^2−^ and four fold CO_3_^2−^ did not have any significant effect on the stripping peak current of Pb^2+^ and Cu^2+^	[74]

* SSA/MoS_2_: 5-sulfosalicylic acid/MoS_2_. PANI-GGHH: polyaniline functionalized with peptide chain glycine-glycine-histidine-histidine. PANI: polyaniline. GO: graphene oxide. NP: nanoparticles. 1 μM Cu(II) = 64 ppb Cu(II).

#### 3.2.7. Detection of Other Metal Ions

As listed in Table 9, the detection of two additional metal ions, Ni(II) and Co(II), has been reported in the literature using SWCNT-FET and chemistor transduction, respectively.

The peptide-functionalized SWCNT-FET mentioned in the previous section for copper detection [113], was also applied to Ni(II) detection with a different peptide sequence. As for Cu(II), a remarkably low LOD was achieved (2.8 ppt) within the range 0.58 to 587 ppt. Such a low LOD is interesting for drink water monitoring as the MAC for Ni(II) is low (20 ppb).

Gou et al. [165] compared flexible polyazomethine -PAM- polymer and rigid (shape persistent macrocycle) functional probes on SWCNTs for chemiresistive Co(II) sensing. They indicate that the flexibility of the PAM allows for better performances as it rearranges over the SWCNT network when binding the metal ions, enabling strong electronic interaction with SWCNT. They report 0.04 ppt of LOD over an extremely large range (0.04 ppt~440 ppm), which is remarkable not only for chemistors (usually less sensitive than FET and electrochemical sensors) but also for electrochemical detection of heavy metals as discussed in the previous sections. It raises the question whether even better LOD could be achieved with alternative transduction modes. 

#### 3.2.8. Multiplexed Detection of Metal Ions

As reported above, studies on metal ions detection rely heavily on electrochemical transduction (32 papers out of 36 in total). Electrochemical detection, and more specifically stripping voltammetry, is particularly interesting for the simultaneous detection of different metals in water, as the current peaks for each metal appear at different voltage range, as can be shown in Figure 7 [129] (obtained with MWCNTs threads electrodes).

Naturally, among these 32 references reporting electrochemical transduction, 14 are reporting on multiplexed detection with stripping voltammetry while none of the four papers based on electrical transduction does. Investigated groups of metal ions are Cd(II)/Pb(II) (7), Cd(II)/Zn(II)/Pb(II) (3), Cd(II)/Zn(II)/Pb(II)/Cu(II) (1), Cd(II)/Cu(II) (1), Cu(II)/Pb(II) (1) and Hg(II)/Cu(II) (1). The simultaneous detection of lead and cadmium is particularly focused on (10 papers out of the 14), as these two heavy metals are commonly found together in soil and water supplies and are both severe environmental contaminants even at trace levels. Table 10 provides a comparison of the performances of the devices reported in these 14 papers as a function of the target species, with conversion from ppb to M (Molar concentration) unit to allow comparison between analytes. Overall, devices have slightly better detection limit to Pb(II) than to Cd(II) irrespective of the functionalization, except for the Bismuth-reduced graphene oxide functionalization reported in [146] with sensitivity to Pb(II) enhanced by a factor of 50 compared to Cd(II). By contrast, the limit of detection to Cd(II) is much lower than that to Zn(II) (by a factor of 6 to 30), except in [129] with non-functionalized MWCNT threads where it is 1.6 times higher. Finally, the limit of detection to Cu(II) is much higher than for other species by about one order of magnitude.

**Table 9 sensors-22-00218-t009:** CNT-based sensors for detecting Ni(II) and Co(II) ions in water.

Type of CNT	Functional Probe	Functionalization	Analyte (Add. Analytes)	Detection Limit	Sensitivity	Transduction Method	Deposition Method	Electrode Material Contact Configuration	Substrate	Interference Study	Ref.
SWCNT	Polypyrrole-His_n_ *	Non covalent	Ni(II)	2.8 ppt	1.5 µS/decade (5%/decade) (0.59 ppt~59 ppb)	FET (liquid gate)	CVD	300 nm Au Pt wire (Counter electrode), Ag/AgCl (Reference electrode)	Si/SiO_2_(120 nm)	His_6_ shows higher chelation power for Ni(II) than to Cu(II).	[113]
SWCNT	PAM *	Non covalent	Co(II)	0.04 ppt	0.014 */decade (0.04 ppt~440 ppm)	Chemistor	Spay-casting	Al tape Ag paint	Si/SiO_2_	Selectivity to Co(II) was investigated in presence of Cu(II). The electrical response was higher with Co(II).	[165]

* His: peptide histidine. PAM: polyazomethine.

**Table 10 sensors-22-00218-t010:** Comparison of the performances of sensors based on multiplexed detection as a function of the target species.

Type of CNT	Functional Probe	Functionalization	Cd(II) LOD	Pb(II) LOD (LOD Pb/Cd)	Zn(II) LOD (LOD Zn/Cd)	Cu(II) LOD (LOD Cu/Cd)	Hg(II) LOD	Ref.
MWCNT	Nafion/Bismuth	Non covalent	0.04 ppb–0.4 nM	0.025 ppb–0.12 nM 0.3				[73]
MWCNT	Bismuth	Non covalent	0.04 ppb –0.4 nM	~0.04 ppb–0.2 nM 0.5				[143]
MWCNT	PSS-Bi	Non covalent	0.02 ppb–0.2 nM	0.04 ppb–0.2 nM 1				[142]
MWCNT	rGO-Bi	Non covalent	0.6 ppb–50 nM	0.2 ppb–1 nM 0.02				[146]
MWCNT	PPy-Bi	Non covalent	0.16 ppb–1.4 nM	0.1 ppb–0.5 nM 0.4				[145]
MWCNT	Fe_3_O_4_-LSG-CS-Bi	Non covalent	0.1 ppb–0.9 nM	0.07 ppb–0.3 nM 0.3				[144]
MWCNT	Sb_2_O_3_	Non covalent	17 ppb–0.15 µM	24 ppb–110 nM 0.7				[136]
MWCNT	Pristine	Non functionalized	8.4 ppb–75 nM	6.6 ppb–31 nM 0.4	28 ppb–0.43 µM 6			[131]
MWCNT	Bismuth	Non covalent	0.8 ppb–7 nM	0.2 ppb–1 nM 0.14	11 ppb–0.17 µM 24			[138]
MWCNT	Bismuth	Non covalent	0.7 ppb–6 nM	1.3 ppb–6.2 nM 1	12 ppb–0.18 µM 30			[139]
MWCNT	Pristine	Non functionalized	0.23 ppb–2 nM	0.3 ppb–1 nM 0.5	0.08 ppb–1.2 nM 0.6	17 ppt–0.26 nM 0.13		[129]
MWCNT	Poly(1,2-diaminobenzene)	Non covalent	0.25 ppb–0.22 nM			0.33 ppb–5 nM 22		[69]
MWCNT	Cysteine	Covalent		1 ppb–4 nM		15 ppb–0.23 µM		[74]
MWCNT	PANi-Bi NPs@GO	Non covalent				32 ppt–0.5 nM	2 ppt–0.01 nM	[151]

PSS-Bi: Poly(sodium 4-styrenesulfonate)-Bismuth. rGO-Bi: Reduced graphene oxide-Bismuth. PPy-BiNPs: Polypyrrole-Bismuth. LSG-Cs-Bi: laser scribed graphene-chitosan-Bismuth. Sb_2_O_3_: antimony oxide. PANI: polyaniline. GO: graphene oxide. NP: nanoparticles. 1 μM Cd(II) = 112 ppb Cd(II). 1 μM Zn(II) = 65 ppb Zn(II). 1 μM Pb(II) = 210 ppb Pb(II). 1 μM Cu(II) = 64 ppb Cu(II). 1 μM Hg(II) = 200 ppb Hg(II).

#### 3.2.9. Interference Studies

Interferent studies are particularly significant regarding to toxic metal ions detection because in most water matrices, a wide range of ions are present at the same time, some of these at concentrations orders of magnitude larger than the target trace metals. For these reasons, most studies include interfering studies (30 out of 36 papers).

Most papers study interferences by other toxic ions typically present in the ppb range in water, such as Cu(II), Fe(II), Fe(III), Ni(II), Zn(II), Cr(III), As(III), Sb(III), Se(IV), Pb(II), Al (III), Fe (III), Ni (II), Co (II), F^−^ and SCN^−^. Among these, Cu(II) is the one reported most consistently as being an interferent for bismuth-functionalized CNT sensors due to the competition between bismuth ions and copper ions. It notably impacts performances for Zn(II), Cd(II), Pb(II) and As(III) detection [132,139,144,145,146,159].

Other papers rather focus on more ubiquitous ions usually present in the ppm range in water, such as Cl^−^, PO_4_^3−^, SO_4_^2−^, NO_3_^−^, Na^+^, Ca(II), Mg(II), K^+^ or CO_3_^2−^ [73,141]. For these ions, no interference to Pb(II), Cu(II) and Cd(II) detection was found.

These two types of interferents are mentioned in the literature on Hg(II) sensors. Out of eight papers, seven report on interferent studies. All these studies conclude toward a strong selectivity toward Hg(II) against the various interferent ions (see Table 10), irrespective of the types of functionalization (pristine, covalent, non-covalent) and of transduction (electrochemical, FET, chemistor). It suggests a strong selectivity of the CNTs themselves toward Hg(II).

Among other chemicals tested for interference, EDTA was found to particularly affect the detection of Cu(II) because EDTA forms complexes with every cation through its two amine and four carboxylate groups [163]. Benzene, xylene and some surfactants also interfere with metal ions detection by preventing the stripping of trace metals during stripping voltammetry measurements.

### 3.3. Nitrogen (Ammonia, Nitrite, Nitrate)

Ammonia (NH_3_) is highly soluble in water and found under the form of dissolved gas or as the ammonium ion (NH_4_^+^) depending on pH. Though it may be present in water as a result of normal biological degradations of proteins, it may also be brought by industrial water discharge. It is also sometimes used for drink water treatment (notably in the USA).

Nitrite ions (NO_2_^−^) are widely used as fertilizing agents and food preservatives. They are in consequence among the pollutants most often identified in natural waters. They are highly toxic for human beings (fatal dose of nitrite ingestion is between 8.7 and 28.3 μM) [166]. Nitrate ions (NO_3_^−^) are also widely found in groundwater and subsequently in drinking water. They primarily result from fertilizers, septic systems and manure storage or spreading operations. Although nitrite, nitrate and ammonia all have strong health and environmental impacts, only nitrite sensing has been reported with CNTs so far. Table 11 summarizes the reported performances. All papers rely on electrochemical transduction with non-covalently functionalized CNTs, and only one study reports the use of SWCNT.

The reported LODs vary from 0.016 µM to 25 µM (1 mM nitrite = 46 ppm), and the ranges of detection cover the scale from 0.1 µM to 10 mM. The MAC for nitrite in drinking water is around 1 ppm/20 µM, so nine papers out of 10 show acceptable limit of detection for nitrite monitoring in drink water with seven papers out of 10 reporting negligible interferences.

The best result is reported with a LOD of 0.016 µM with a functional probe based on a nanocomposite made of Co_3_O_4_ and rGO (reduced graphene oxide) [167]. With the same electrode and deposition process (drop casting on glassy carbon, using rGO only as the functional probe leads to a considerably higher LOD of 25 µM [76], underlying the role of the cobalt oxide functionalization in the sensitivity. Consistently, cobalt oxide on its own has been reported to be promising for nitrite sensing by its reduction process upon exposure to nitrite [168]. 

**Table 11 sensors-22-00218-t011:** CNT-based nitrogen sensors for water quality monitoring, sorted by detection limit.

Type of CNT	Functional Probe	Functionalization	Analyte	DETECTION LIMIT	Sensitivity (Detection Range)	Transduction Method	Deposition Method	Electrode Material Contact Configuration	Substrate	Interference Study	Ref.
MWCNT	Co_3_O_4_^−^ rGO *	Non covalent	Nitrite	0.016 µM	0.408 µA/µM/cm^2^ (0.1~8000 µM)	Voltammetry	Drop casting	Glassy carbon electrode	Glassy carbon	100-fold of alcohol, Na^+^, K^+^, Cl^−^, NO_3_^−^, N_2_H_4_, SO_3_^2^^−^,SO_4_^2^^−^, has no effect on sensor response.	[167]
MWCNT	PCMA *	Non covalent	Nitrite	0.067 µM	−0.023 µA/µM (1~10 µM) −0.022 µA/µM (10~100 µM) −0.034 µA/µM\ (100~1000 µM) −0.026 µA/µM (1000~4000 µM)	Differential pulse voltammetry, Chronoamperometry	Drop cast of PCMA/MWCNT, then electrochemical crosslinking	Au	Au	Not provided	[169]
MWCNT	AuNPs/ PEI */ MWCNT-COOH	Non covalent	Nitrite	0.2 µM	−0.500 µA/µM * (1~2000 µM) −58 µA/mM (1~1400 µM)	Voltammetry	Drop casting	Au	Au	Na^+^, Mg(II), Ca(II), Zn(II), Fe(II), Cl^−^, I^−^ and SO_4_^2−^ did not have significant interference in the detection of nitrite.	[170]
SWCNT	Pd	Non covalent	Nitrite	0.25 µM	420 µA mM^−1^ cm^−2^ (2~240 µM ) 190 µA mM^−1^ cm^−2^ (280~1230 µM)	Differential pulse voltammetry	Vacuum filtration	SWCNT	PET	Negligible effect of K^+^, Na^+^, Cl^−^, PO_4_^3−^, NH_4_^+^, CH_3_COO^−^ and Zn(II) in concentration above500 mM and concentrations of Mg(II), Ca(II), Cd(II), CO_3_^2−^, NO_3_^−^,and SO_4_^2−^ above 200 mM	[171]
MWCNT	Ni_7_S_6_	Non covalent	Nitrite	0.3 µM	0.185 µA/µM (1~4200 µM)	Voltammetry	Drop casting	Glassy carbon electrode	Glassy carbon	Results comparable to high-performance liquid chromatography for lake water, tap water and pickle water	[172]
MWCNT	GO-MWCNT-PMA-Au	Non covalent	Nitrite	0.67 µM	0.484 µA/µM (2~10,000 µM)	Differential pulse voltammetry	Drop casting	Glassy carbon electrode	Glassy carbon	No obvious response was observed when injection of 0.4 Mm of Na^+^, Ca(II), NO_3_^−^, CO_3_^2-^, K^+^, Cl^−^, SO_4_^2−^, IO_3_^−^	[173]
MWCNT	Au/TiO_2_	Non covalent	Nitrite	3 µM	N/A (4~225 µM)	Differential pulse voltammetry	Pulsed electrodeposition	Glassy carbon electrode	Glassy carbon	The presence of arginine, serine, tyrosine, cysteine, glucose, alanine (each of 0.1 mM) causes less than 5% variation on sensor response.	[166]
MWCNT	Thionine	Non covalent	Nitrite	4 µM	0.002 µA/µM (6 µM~15,000 µM)	Voltammetry	Transfer via abrasion from filter paper to heated GC electrode	Glassy carbon electrode	Glassy carbon	Not provided	[174]
MWCNT	PANI *	Non covalent	Nitrite	6.1 µM	0.684 µA/µM/cm^2^ (N/A)	Voltammetry	Electrodeposition	Glassy carbon electrode	Glassy carbon	Not provided	[175]
MWCNT	rGO *	Non covalent	Nitrite	25 µM	0.01 µA/µM (75~6060 µM)	Differential pulse voltammetry	Drop casting	Glassy carbon electrode	Glassy carbon	0.6 M Ca(II), Cu(II), K^+^, Na^+^, Zn(II),SO4^2−^, l-cysteine, NO_3_^−^ and Cl^−^ did not interfere with the pick signals of 0.15 mM HQ, 0.15 mM CC, 0.15 mM PC and 0.15 mM NO_2_^−^.	[76]

* In case of Nitrite ion (NO_2_^−^), 1 mM = 46 ppm. PEI: polyethyleneimine. NP: nanoparticles. PCMA: poly(VMc-co-VCz-coAA; VMc: 7-(4-vinylbenzyloxy)-4-methyl coumarin, VCz: 9-Vinylcarbazole, AA: Acrylic acid), GO: Graphene oxide; rGO: Reduced graphene oxide, GCE: Glassy carbon electrode, PANI: Polyaniline.

### 3.4. Water Hardness (Ca(II), Mg(II))

In general, the total hardness of water is defined as the sum of the concentrations of divalent calcium and magnesium, and of all the other alkaline earth ions in the water matrix. The concentration of calcium and magnesium ions is dominant to the other alkaline-earth metals, therefore water hardness is generally estimated from the concentration of these two ions [176]. Determination of water hardness is important as hard water can precipitate inside a water pipe and cause limescale. The sum of recommended Ca(II) and Mg(II) concentration in water ranges from 2 to 4 mM.

There have been relatively few studies on CNT sensors for Ca(II) and Mg(II) ions measurement, as reported in Table 12, all addressing functionalized CNT. To be noted, both sensors are tested to measure either Ca(II) or Mg(II) concentration in water, not the total water hardness. The best reported limit of detection is achieved with a chemFET approach, reaching down to 100 pM of Ca(II) (4 ppt). It is based on the functionalization of SWCNT by Fluo-4 AM (Fluorescent acetoxymethyl ester). It is a fluorescein derivative comprising amino carboxylate coordinating groups that has been widely used for calcium detection [177].

**Table 12 sensors-22-00218-t012:** CNT-based water hardness sensors in water.

Type of CNT	Functional Probe	Functionalization	Analyte	Detection Limit	Sensitivity (Detection Range)	Transduction Method	Deposition Method	Electrode Material Contact Configuration	Substrate	Comments	Ref.
SWCNT	Fluo-4 AM *	Non-covalent	Ca(II)	100 pM	69 nA/decade (100 nM~1 mM)	CNT-FET	Dip coating	Ti (10 nm)/Au (30 nm) (liquid, floating gate)	Glass (borosilicate glass capillary)	FET at the end of a nanoneedle for intracell monitoring	[178]
MWCNT	PDMS *	Non-covalent	Ca(II) (Mg(II))	25 µM	N/A (25 µM~5 mM (Not linear))	Capacitive measurement	Mold injection and thermal curing	MWCNT	PDMS	Measured at 2.4 kHz frequency	[179]

* Fluo-4 AM: Fluorescent acetoxymethyl ester. PDMS: Polydimethylsiloxane. 1 ppm Ca(II) = 0.025 mM Ca(II).

### 3.5. Dissolved Oxygen (DO)

Dissolved oxygen (DO) refers to the amount of free oxygen present in water in gaseous form. It is measured in mg/L or in ppm. Algal biomass, dissolved organic matter, ammonia, volatile suspended solids and sediment oxygen demand can affect the variation of DO in water. Hence DO is widely used as indicator of the metabolism and pollution levels of waterbodies [180,181].

Several groups reported that molecular oxygen acts as dopant for CNTs and thus limits the selectivity and sensitivity of CNT-based sensors to other gases (in air) or chemicals (in water) [182,183]. In turn, this suggested the feasibility of CNT-based DO sensors. Table 13 shows the two reported DO sensors based on CNT reported so far. Both are based on cyclic voltammetry with non-covalently functionalized MWCNT coated on glassy carbon electrodes.

Regarding the first reported CNT-based dissolved oxygen sensor in 2004 [184], the functional probe is hemin. Hemin is an iron-containing porphyrin that can be found in red blood cells, and that efficiently binds dioxygen [185]. Hemin-functionalized MWCNTs show a better sensitivity to O_2_ than non-functionalized ones in O_2_-saturated phosphate buffer solution.

More recently, Tsai et al. (2013 [186]) used gold nanoparticles as functional probes, gold being selected as an effective catalyst for oxygen reduction. The electrodes showed a quasi-linear response to dissolved oxygen with a detection limit at 0.1 ppm (~3 µM). Such resolution is suitable to determine the spatial variation of DO concentration for oxygen profiling in water bodies [187]. 

### 3.6. Disinfectants (Hypochlorite, Hydrogen Peroxide, Chloramine, Peracetic Acid)

Free chlorine, hydrogen peroxide, peracetic acid, potassium permanganate and chloramine, are chemicals with an outstanding oxidation capacity. They are used either in the initial disinfection process of water or to keep the drinking water disinfected during distribution.

One of the most widely used drink water disinfectants is free chlorine. Its concentration in water should be in the range from 0.2 to 0.5 mg/L after disinfection (in which case it is called residual free chlorine). At lower concentrations, bacterial contamination may occur; at higher concentrations it is hazardous to human health.

Table 14 shows the reported CNT-based sensors for detecting disinfectants in water. The detection of hydrogen peroxide is reported in six articles, all but one by the means of electrochemical measurements with non-covalently functionalized CNT. The detection limit is about 3 ppm for the chemistor device (which is acceptable for the applicative range: conventional hydrogen peroxide sensors have a range from 0 to 2000 ppm and EPA (US) recommended levels in drink to range from 25 to 50 ppm). The use of electrochemical transduction with CNT functionalized by metallic materials lowers this threshold by several orders of magnitude, reaching down to 3.4 ppb with a 3D structure based on nitrogen doped Co-CNTs over graphene sheets [188]. An approach based on petal-like chromium hexacyanoferrate (Cr-hcf) crystallites yielded 17 ppb detection limit, this later material being specifically studied because of its electrocatalytic activity in the reduction of H_2_O_2_ [75]. As in the previous sections, the use of these two types of 3D structuration appeared to lead to large improvement (more than one order of magnitude) of performances compared to more traditional 2D architecture such as [189,190] (also metal based).

Regarding free chlorine (or hypochlorite detection for detection at pH higher than 7), the non-functionalized, aligned MWCNT-based chemistor device shows the lowest LOD below 5 ppb. The sensitivity in this reference is attributed to the oxidative properties of NaOCl leading to doping effect of the CNTs [191].

**Table 13 sensors-22-00218-t013:** CNT-based dissolved oxygen sensors in water.

Type of CNT	Functional Probe	Functionalization	Analyte	Detection Limit	Sensitivity (Linear Range)	Transduction Method	Deposition Method	Electrode	Ref.
MWCNT	Hemin	Non-covalent	O_2_	N/A	N/A (N/A)	Cyclic voltammetry, Amperometry	In-place CVD (densely-packed, vertically aligned CNTs)	Glassy carbon electrode	[184]
MWCNT	Au NP *	Non-covalent	O_2_	0.1 ppm	N/A (0~50 ppm)	Cyclic voltammetry	Not provided	Glassy carbon electrode	[186]

* NP nanoparticles.

**Table 14 sensors-22-00218-t014:** CNT-based sensors for detecting disinfectants in water. References are sorted by type of analyte (hydrogen peroxide, free chlorine) then by limit of detection.

Type of CNT	Functional Probe	Functionalization	Analyte	Detection Limit	Sensitivity (Detection Range)	Transduction	Deposition Method	Electrode Material Contact Configuration	Substrate	Interference	Ref.
MWCNT	PVC, DBE *	Non covalent	Hydrogen peroxide	N/A	Not Provided	Amperometry, voltammetry	Screen Printing	CNT electrodes	Alumina	Not provided	[192]
MWCNT	Nitrogen doped Co-CNTs over graphene sheets	Non covalent	Hydrogen peroxide	100 nM 3.4 ppb	−0.85 µA/ppm	Voltammetry, amperometry	Coating	Glassy carbon electrode	Glassy carbon electrode	No interference with uric acid, ascorbic acid and glucose	[188]
SWCNT	Cr-hcf *	Non covalent	Hydrogen peroxide	0.5 µM 17 ppb	1 µA/ppm (17 ppb~340 ppm) *	Amperometry, voltammetry	Drop casting	Glassy carbon electrode	Glassy carbon	No interference from ascorbic acid and uric acid	[75]
CNT (probably Multi-walled)	Fe-Ni	Non covalent	Hydrogen peroxide	16 µM 540 ppb	1.2 µA/ppm (34 ppm~510 ppm)	Voltammetry	Paste poured into electrode	Glassy carbon electrode	Glassy carbon electrode	Not provided	[189]
MWCNT	Chitosan/Cu/MWCNT-COOH	Non covalent	Hydrogen peroxide (pH)	<25 µM <850 ppb	0.97 nA/ppb (500 µM~10 mM)	Amperometry	Potentiostatic polarization	Glassy carbon electrode	Chitosan-coated glassy carbon	No interference from ascorbic acid and uric acid	[190]
SWCNT	Phenyl capped aniline tetramer	Non covalent	Hydrogen peroxide	<3 ppm	1%/ppm (3 ppm~8 ppm) Nonlinear <1%/100 ppm (48 ppm~1200 ppm)	Chemistor	Drop casting	Carbon ink	Glass	Not provided	[193]
MWCNT	Pristine	Non functionalized	Free chlorine in its hypochlorite ion form	<5 ppb	Logarithmic 39% /decade * (0.03~8 ppm)	Chemistor	Dielectrophoresis (aligned MWCNT)	Cr/Au	Glass	No information about selectivity, pH information not provided	[191]
MWCNT	Epoxy EpoTek H77A	Non covalent	Free chlorine under hypochlorous acid form (At pH 5.5)	20 ppb	0.15 µA/ppb (0.02~4 ppm)	Voltammetry	Paste poured into tube and thermally cured	Epoxy/MWCNT composite	Not provided (tube)	Validated in real water matrices (tap water and swimming pool)	[194]
SWCNT	Phenyl capped aniline tetramer	Covalent	Free chlorine	<60 ppb	92 nA/decade (0.06~60 ppm (linear up to 6 ppm))	Chemistor	Drop casting	Au	Glass	Non selective to different oxidants list of oxidants not provided Regeneration possible	[53]

* Cr hcf: Chromium hexacyanoferrate. PVC: Polyvinyl chloride. DBE: dibasic ester. 1 mM = 34 ppm H_2_O_2_.

### 3.7. Sulfur (Sulfide, Sulfite, Sulfate)

Sulfur can be found in aqueous environments in oxidized form as sulfite (SO_3_^2−^), sulfate (SO_4_^2−^), or in reduced form as sulfide (S^2−^). The oxidized forms of sulfur play an important role within environmental systems [195]. In fact, they are detected in natural waters, waste waters and in boiler waters (those treated with sulfur for dissolved oxygen control). High concentrations of sulfite in boiler waters is harmful, since it decreases pH and subsequently, stimulates corrosion.

Table 15 compares the different CNT-based sulfur (sulfite and sulfide) sensors used for water quality monitoring. No sulfate sensor has been reported yet. All of these studies address electrochemical sensing with non-covalently functionalized CNTs.

Regarding sulfite detection, both reports [68,78], use a functional probe based on ferrocene. Zhou et al. (2008) used ferrocene-branched chitosan composites, while Hassan et al. (2011) used only ferrocene for GCE modification. Indeed, ferrocene and its derivatives have been reported as strong electrocatalysts for sulfite detection [196]. The LOD and sensitivity of the probe using ferrocene being directly in contact with MWCNTs are better by a factor of more than 20 than those of the probe using ferrocene-branched chitosan.

Regarding the detection of sulfide, all reports address electrochemical sensing with MWCNTs. Best LODs are in the range 0.2 to 0.3 µM (1 mM sulfide = 34 ppm). These LOD are too high compared to drink water quality requirements as sensitivity to sulfide in the ppt to sub-ppm range is required. The best limit of detection of 0.2 µM is reported with Hematoxylin [197], a compound reported to foster electrocatalytic oxidation of sulfide. Platinum [198] nanoparticles (also expected to oxidize sulfide) electrodeposited on vertically aligned CNT arrays perform also very well, comparably to non-functionalized CVD-grown MWCNTs [199]. 

**Table 15 sensors-22-00218-t015:** CNT-based sulfur sensors for water quality monitoring. References are sorted by analyte (sulfite and sulfide), then by limit of detection.

Type of CNTs	Functional Probe	Functionalization	Analyte	Detection Limit	Sensitivity (Detection Range)	Transduction Method	Deposition Method	Electrode Material Contact Configuration	Substrate	Interference Study	Ref.
MWCNT	Ferrocene-branched chitosan	Non covalent	Sulfite	2.8 µM	0.013 µA/µM (5 µM~1500 µM)	Amperometry	Drop casting	Glassy carbon electrode	Glassy carbon	600-fold excess of Ca(II), Mg(II), Ba(II), PO_4_^3−^, NO_3_^−^, CO_3_^2−^ and Cl^−^ did not interfere in the determination of sulfite.	[68]
MWCNT	Ferrocene	Non covalent (Physical immobilization)	Sulfite	0.1 µM	3.3 µA/µM (0.4 µM~4 µM) 0.18 µA/µM (4 µM~120 µM)	Differential Pulse Voltammetry	Paste mixture with graphite powder blended with paraffin oil	MWCNT paste, Copper wire	Glass tube	Not provided	[78]
MWCNT	Hematoxylin	Non covalent	Sulfide	0.2 µM	103 nA/µM (0.5 µM~150 µM)	Amperometry	Paste mixture of MWCNT, mineral oil and graphite powder	Carbon paste	Teflon tube	No interference with Sn(II), Co(II), (II)Pb(II), (II)Zn(II), Cu(II), Ni(II), Mn(II), Fe(II) and Fe(III)	[197]
MWCNT	Platinum	Non covalent (plating)	Sulfide	0.26 µM	0.63 µA/µM (0.26 µM~40 µM and 40 µM~100 µM)	Amperometry Differential pulse voltammetry	Thermal CVD (vertically aligned CNTs)	Stainless steel	Stainless steel	Not provided	[198]
MWCNT	Pristine	Not functionalized	Sulfide	0.3 µM (CVD *), 12.5 µM (ARC *)	0.12 µA/µM (1.3 µM~113 µM) (CVD), 0.005 µA/µM (12.5 µM~87.5 µM) (ARC)	Hydrodynamic voltammetry	Drop casting	Glassy carbon electrode	Glassy carbon	Not provided	[199]
MWCNT	Copper phenanthroline	Non covalent (Physical immobilization)	Sulfide	1.2 µM	34 nA/µM (5 µM~400 µM)	Amperometry	Drop casting	Glassy carbon electrode	Glassy carbon	No interference with SO_3_^2−^, SO_4_^2−^, S_2_ O_3_^2−^, S_4_ O_6_^2−^, Cysteine.	[200]

* ARC: Arc discharge method, CVD: Chemical vapor deposition method.

### 3.8. Other Contaminants

The detection of various additional analytes is also reported in the literature, as detailed in Table 16.

Zhao et al. reported that the threshold voltage of a CNT-FET with interdigitated electrodes using pristine, in-place grown SWCNT showed a response to glycerol in water [101]. This response is attributed to polar glycerol molecules adsorbing on the SWCNT sidewalls and acting as dopant for SWCNT. Glycerol is relevant to monitor in water as it is widely used in the food, beverage and e-cigarettes industry, and is also used in the formulation of numerous solvents. Thus, it ends up in the water cycle from human and industrial waste and may feature ecotoxicity [201].

Regarding to security applications (detection of explosive materials at extremely low concentration in water), Wei et al. demonstrated that a SWCNT-based chemistor functionalized with 1-pyrenemethylamine (PMA) could detect 2,4,6-trinitrotoluene in water, with a detection limit of 10 ppt and less than 1 min of response time [55]. The sensor showed high selectivity to several interfering molecules, for example, 2,6-dinitrotoluene (DNT) and 2,4-dinitrotoluene (2,4-DNT). The amino substituent in PMA was reported to interact selectively with TNT by forming negatively-charged complexes on the SWCNT sidewalls.

Regarding to the identification of dangerous toxins, Lee et al. reported that CNT-FET showed a response to botulinum neurotoxin (BoNT) in water with a detection limit up to 60 pM in case of peptide-modified CNT ((A)), and 52 fM in case of CNT modified with the anti-botulinum neurotoxin (B)) [109].

The detection of coliforms (notably *Escherichia coli*, but also other bacterial pathogens) is of major impact to drink water quality monitoring. However, standard assays take 24 h to 48h to determine presence or absence of coliforms, so reducing this detection time is of major interest. It relies on indirect detection of chemicals released by the bacteria, often upon addition of reagents. In [202], *p*-aminophenol is used as an indicator of the presence of coliform and detected through a glassy carbon electrode coated with Nafion/MWCNT. Coliform detection down to 10 cfu/mL was possible with 5 h response time.

Finally, references [76,203,204,205] address the topic of emerging contaminants with electrochemical sensors based on CNT modified carbon electrodes, through the angle of drugs and hormones [76,203] and of bisphenol A [204,205,206]. Emerging contaminants are compounds derived from manufactured chemicals and that despite being present only in µg/L concentrations (or below) in water bodies are known to have strong impact on health and environment [207]. Among these, bisphenol A is notably acknowledged as endocrine disruptor and as toxic to reproduction. It is worth mentioning that regarding drug and health-care related chemicals, there are more references available beyond the field of drink water monitoring which are not included here, as detailed recently in [208]. 

**Table 16 sensors-22-00218-t016:** Reported chemical CNT sensors for water quality monitoring with different probes and analytes.

**Type of CNT**	**Functional Probe**	**Functionalization**	**Analyte**	**Detection Limit**	**Sensitivity** **(Detection Range)**	**Transduction Method**	**Deposition Method**	**Electrode Material** **Contact Configuration**	**Substrate**	**Interference Study**	**Ref.**
SWCNT	Pristine	Not functionalized	Glycerol	N/A	~10 Ω/Glycerol by weight % in water (10~50%)	CNT-FET	Dielectrophoresis	Cr/Au	Si/SiO_2_	Not provided	[101]
SWCNT	1-phyrenemethylamine	Non-covalent	Trinitrotoluene	~ppt	N/A (>0.01 ppb)	Chemistor with interdigitated electrodes (IDEs)	Dip coating	Cr/Au	Si/SiO_2_	Relatively selective to 2,6-DNT *, 2,4-DNT, 1,3-DNB *, 1-NB *, Response time~30 s	[55]
SWCNT	Peptides, anti-BoNT/E-Lc *	Non-covalent	BoNT*	60 pM (Peptide probe), 52 fM (Anti-BoNT/E-Lc probe)	27.95 nS/nM (Peptide), 313 nS/pM (Anti-BoNT)	CNT-FET	CVD (vertically aligned SWCNTs)	Au foils Bottom gate	120 nm SiO_2_ on PDMS film	Not provided	[109]
MWCNT	Nafion	Non covalent	p-aminophenol (Coliforms)	10 cfu/mL	10 to 10^4^ cfu/mL	Cyclic voltammetry, amperometry	Drop casting	Glassy carbon electrode	Glassy carbon	Not provided	[202]
MWCNT	rGO*	Non covalent	Hydroquinone Catechol p-cresol (nitrite)	2.6 µM 1.8 µM 1.6 µM	0.19 µA/µM (8~391 µM) 0.07 µA/µM (5.5~540 µM) 0.04 µA/µM (5~430 µM)	Differential pulse voltammetry	Drop casting	Glassy carbon electrode	Glassy carbon	0.6 M Ca(II), Cu(II), K^+^, Na^+^, Zn(II),SO4^2−^, l-cysteine, NO_3_^−^ and Cl^−^ did not interfere with the pic signals of 0.15 mM HQ, 0.15 mM CC, 0.15 mM PC	[76]
MWCNT	Fe-Co doped TNTs	Non-covalent	Sulpiride	87 nM	58.8 mV/decade (100 nM~10 mM)	Potentiometry	Paste mixture of graphite powder, MWCNT, Fe-CO-TNT, βCD ionophore, NaTPB anionic additive, DBP plasticizer	Carbon paste electrode	Syringe	No interference observed with K^+^, Na^+^, Ca(II), Mg(II), Cd(II), Co(II), Mn(II), Fe(II)	[203]
SWCNT	βCD *	Covalent	Bisphenol A	1.0 nM	1.3 mA/mM 11 nM–19 µM	Cyclic voltammetry	Drop casting	Glassy carbon electrode	Glassy carbon	No interference study, but tested on real plastic samples	[206]
MWCNT	βCD	Covalent	Bisphenol A	14 nM	7.2 µA/µM (125 nM~2 µM) 2.2 µA/µM (2 µM~30 µM)	Linear sweep voltammetry	Drop casting	Screen printed carbon electrode	Not provided	Selective to APAP *, BPA *, BPS	[204]
MWCNT	ZIF-67 *	Covalent	TBBPA *	4.2 nM	21.08 µA/µM (0.01~1.5 µM)	Differential pulse voltammetry, cyclic voltammetry	Paste mixture of paraffin oil, AB * and CNTs	Carbon paste electrode	Syringe	TBBME *, TBBDE *, BPAF *, BPA *, TCBPA *, TBBPS * did not show remarkable interference.	[205]

* DNT: Dinitrotoluene. DNB: Dinitrobenzene. NB: Nitrobenzene. BoNT: Botulinum neurotoxin. rGO: reduced graphene oxide. TNT: titanate nanotube. E-Lc: E light chain. ZIF-67: Zeolitic imidazole framework-67. βCD: β-cyclodextrin. TBBPA: Tetrabromobisphenol A. AB: acetylene black. DBP: dibutyl phthalate. NaTPB: sodium tetraphenylborate. BPA: bisphenol A. TBBME: tetrabromobisphenol A-bis(dibromopropyl ether). TCBPA: tetrachlorobisphenol A. BPAF: hexa-fluorobisphenol A (BPAF, 98%), TBBPS: 4,4-sulphonyl-bis-(2,6-dibromophenol). TBBDE: tetrabromobisphenol A diallyl ether. BPS: bisphenol S. APAP: acetaminophen.

## 4. Discussion

### 4.1. Summary of Best Performances

The previous paragraphs show the various performances of the CNT sensors for all the reported analytes. To get an overview on the best device strategy to achieve the best performances, Table 17 summarizes the references for each type of functionalization and transduction for the 15 analytes that are addressed by more than one reference: pH, Pb(II), Cd(II), Zn(II), Hg(II), As(III), Cu(II), nitrite, calcium(II), dissolved oxygen, hydrogen peroxide, free chlorine, sulfite and sulfide. 

### 4.2. Discussion on Sensor Design Choices

#### 4.2.1. Choice of Transduction Mode

There are five analytes for which different transduction modes may be compared: pH, Hg(II), Cu(II), Ca(II) and H_2_O_2_:For pH, FET and impedance spectroscopy reach the same performance and are only slightly better than chemistor.For Cu(II), the LOD achieved with FET is three times better than voltammetry.For Hg(II), the LOD achieved with voltammetry is three orders of magnitude better than that obtained with FET, the latter being two orders of magnitude better than with chemistor.For H_2_O_2_, the LOD achieved with voltammetry is three orders of magnitude better than with a chemistor.For Ca(II), the LOD achieved with FET is five orders of magnitude better than capacitive measurements (which can be seen as a derivative of impedance spectroscopy).

While electrochemical measurements have been more widely used than FET-based approaches (probably due to easier manufacturing), the latter reach comparable or even widely improved performances for three out of four analytes. Testing FET architectures on a wider range of analytes would thus be valuable, as FETs are expected to be easier to operate than electrochemical sensors in field conditions.

Regarding chemistors, they feature larger limits of detection than the two other types, but the comparison is only possible on three analytes (out of 15). Moreover, for two out of three of these analytes (pH and H_2_O_2_), the detection limits are still acceptable for the monitoring of drink water. Finally, for several analytes (Co(II) and trinitrotoluene), ppt level detection limits are possible with chemistors. Considering that chemistors are easier to fabricate than FET and to operate than electrochemical sensors, their extensive testing against other types of analytes would be useful as well.

#### 4.2.2. Functionalized versus Non-Functionalized CNT

Comparison of limits of detection between functionalized and non-functionalized CNTs (including COOH-CNTs) is possible for nine analytes. For Pb(II), Cd(II), Hg(II), As(III) and Cu(II), the use of functionalization improves significantly the limit of detection. Non-functionalized or COOH-functionalized CNT provide best performances for pH, free chlorine and Zn(II).

This suggests to systematically compare the performances of non-functionalized and functionalized CNTs (with the same device architecture), as they may be very sensitive without functionalization.

**Table 17 sensors-22-00218-t017:** Summary of best performances for all analytes addressed by more than one reference. When several transduction types or functionalization strategies are available for a given analyte, the table includes the best performing reference for each type.

Analyte (Add. Analytes)	Type of CNT	Functional Probe	Functionalization	Detection Limit	Sensitivity (Detection Range)	Transduction Method	Deposition Method	Electrode Material Contact Configuration	Substrate	Interference Study	Ref.
pH	MWCNT	Pristine	Non functionalized	N.P.	63 Ω/pH 18%/pH pH 5~9	Chemistor	Sucked by vacuum force	MWCNT	Filter paper	Not provided	[123]
	**SWCNT**	**Pristine**	**Non functionalized**	**1 mM**	**7600 mV/pH** **23%/pH** **(Dual-gate mode)** **pH 3~10**	**FET, potentiometry** **(double gate)**	**Spin coating**	**100 nm Ti contacts for source, drain and top gate**	**p-Si (substrate acting as bottom gate)**	**Not provided**	**[54]**
	**SWCNT**	Poly(1-aminoanthracene)	Non covalent	1 μM	FET 19 µS/pH 14%/pH	FET, potentiometry (liquid gate)	Dielectrophoresis (aligned CNTs)	Au contacts, Pt wire (Auxillary), Ag/AgCl electrode (Reference)	Si/SiO_2_ (300nm)	Multiplexed detection of Ca(II) and Na^+^	[60]
	**MWCNT**	**COOH**	**Covalent**	**N.P.**	**17 Ω/pH** **23%/pH** **(Au)** **pH 4~9**	**Impedance spectroscopy**	**Dip coating**	**Au and Al interdigitated electrodes**	**Kapton^®^**	**Not provided**	**[59]**
Pb(II)	MWCNT	Pristine	Non functionalized	0.3 ppb	2.2 nA/ppb (210~830 ppb)	Stripping voltammetry	CNT thread	Metal wire and silver conductive epoxy	Glass capillary	Simultaneous determination of Cd(II), Cu(II), Pb(II) and Zn(II) demonstrated The presence of Dissolved Oxygen changes the calibration law for Cd(II)	[129]
	**MWCNT**	**Ionic liquid—dithizone based bucky-gel**	**Covalent**	**0.02 ppt**	**0.024 µA/ppb** **(0.1ppt** **~210 ppb)**	**Stripping voltammetry**	**Drop-casting**	**Glassy carbon electrode**	**Glassy carbon**	**No interference of Cd(II) and Cu(II) ions with the detection of Pb(II) ion.**	**[132]**
	MWCNT	Nafion/Bismuth	Non covalent	25 ppt	0.22 µA/ppb (0.05 to 5 ppb) 0.27 µA/ppb (5~100 ppb)	Stripping voltammetry	Drop casting	Glassy carbon electrode	Glassy carbon	500-fold of SCN^−^, Cl^−^, F^−^, PO_4_ ^3−^, SO_4_^2−^, NO_3_^−^ and various cations such as Na^+^, Ca(II), Mg(II), Al(III), K^+^, Zn(II), Co(II) and Ni(II) had no influences on the signals of Pb(II) and Cd(II).	[73]
	MWCNT	PSS-Bi	Non covalent	0.04 ppb	0.079 µA/ppb (0.5~90 ppb)	Stripping voltammetry	Drop casting	Glassy carbon electrode	Glassy carbon	20-fold amounts of Zn(II), 5-fold amounts of Sn(II) and 1-fold amounts of Cu(II) have influence on the determination of Cd(II) and Pb(II) with deviation of 10%.	[142]
Cd(II)	MWCNT	Pristine	Non functionalized	0.23 ppb	3.9 nA/ppb (170~500 ppb)	Stripping voltammetry	CNT thread	Metal wire and silver conductive epoxy	Glass capillary	Simultaneous determination of Cd(II), Cu(II), Pb(II) and Zn(II) demonstrated The presence of Dissolved Oxygen changes the calibration law for Cd(II)	[129]
	**MWCNT**	**PSS-Bi**	**Non covalent**	**0.02 ppb**	**0.23 µA/ppb** **(0.5** **~** **50 ppb)**	**Stripping voltammetry**	**Drop casting**	**Glassy carbon electrode**	**Glassy carbon**	**20-fold amounts of Zn(II), 5-fold amounts of Sn(II) and 1-fold amounts of Cu(II) have influence on the determination of Cd(II) and Pb(II) with deviation of 10%.**	**[142]**
**Zn(II)**	**MWCNT**	**Pristine**	**Non functionalized**	**0.08 ppb**	**3.4 pA/ppb** **(200~590 ppb)**	**Stripping voltammetry**	**CNT thread**	**Metal wire and silver conductive epoxy**	**Glass capillary**	**Simultaneous determination of Cd(II), Cu(II), Pb(II) and Zn(II) demonstrated** **The presence of Dissolved Oxygen changes the calibration law for Cd(II)**	**[129]**
	MWCNT	Bismuth	Non covalent	11 ppb	0.18 µA/ppb (12~18 ppb) 0.24 µA/ppb (20~100 ppb)	Stripping voltammetry	Screen printing	Screen printed MWCNT based electrode	Ceramic substrates	N.P.	[138]
Hg(II)	SWCNT	Pristine	Non functionalized	0.6 ppm	12 mV/ppm (1~30 ppm)	Chemistor	CVD	SWCNT	Glass	1000 fold excess of Fe(II), Fe(III), Ni(II), Cu(II),Zn(II), Cr(III) and 500 folds of As(III), Sb(III), Se(IV) and Pb(II) had no interfering effect in the analysis of mercury solution.	[147]
	SWCNT	Pristine	Non functionalized	2 ppb	0.22/decade 0.2 ppb~201 ppm	FET (Liquid gate)	Dip coating with selective CNT placement	Pd/Au (10/30 nm)	Glass	Good selectivity towards interferent ions	[110]
	SWCNT	Thiophenol	Covalent	0.6 ppb	0.14 µA/ppb (1~18 ppb)	Stripping voltammetry	Dip coating	Au	Au	The presence of 100-fold concentration of Cr(II), Mn(II), Co(II), Ni(II), Zn(II), 50-fold concentration of Fe(II),and 20-fold Cu(II), have no influence on the signals of 50 nM Hg(II) with deviation below 5%.	[150]
	**MWCNT**	**PANi-Bi NPs@GO**	**Non covalent**	**2 ppt**	**1.3 µA/ppb** **(2 ppt~1000 ppm)**	**Differential pulse voltammetry**	**Screen printing**	**(commercial) Carbon ink**	**PET**	**Not provided**	**[151]**
As(III)	MWCNT	COOH	Covalent	N.A	0.24 µA/ppb (0.3~50 ppb)	Stripping voltammetry	Dip coating	Au	Au electrode	Interference was significant when the Sb/As ratio is higher than 1.	[154]
	**MWCNT**	**Au-NP**	**Non covalent**	**0.1 ppb**	**26 µA/ppb** **(75 ppt–5.3 ppm)**	**Stripping voltammetry**	**Drop casting**	**Glassy carbon**	**Glassy carbon**	**Not provided**	**[157]**
**Cu(II)**	**SWCNT**	**PANI-GGHH**	**Non covalent**	**3 ppt**	**N/A** **(3~29 ppt)**	**FET** **(liquid gate)**	**CVD**	**300 nm Au**	**Si/SiO_2_ (120nm)**	**His_6_ shows higher chelation power for Ni(II) than to Cu(II).**	**[113]**
	MWCNT	C_24_H_30_N_6_ Schiff base	Non covalent	10 ppt	N/A (0.09~340 ppb)	Stripping voltammetry	Paste of MWCNT, Schiff base and mineral oil	Copper wire	Filter membrane	Not provided	[160]
	MWCNT	Pristine	Non functionalized	17 ppt	9.4 pA/ppb (32~220 ppb)	Stripping voltammetry	CNT thread aspirated into a glass capillary	Metal wire and silver conductive epoxy	Glass capillary	Simultaneous determination of Cd(II), Cu(II), Pb(II) and Zn(II) demonstrated The presence of Dissolved Oxygen changes the calibration law for Cd(II)	[129]
**Nitrite**	**MWCNT**	**Co_3_O_4_-rGO**	**Non covalent**	**0.016 µM**	**0.408 µA/µM/cm^2^** **(0.1~8000 µM)**	**Voltammetry**	**Drop casting**	**Glassy carbon electrode**	**Glassy carbon**	**100-fold of alcohol, Na^+^, K^+^, Cl^−^, NO_3_^−^, N_2_H_4_, SO_3_^2^^−^, SO_4_^2^^−^, has no effect on sensor response.**	**[167]**
**Ca(II)**	**SWCNT**	**Fluo-4 AM**	**Non-covalent**	**100 pM**	**69 nA/decade** **(** **100 nM~1 mM)**	**FET**	**Dip coating**	**Ti (10 nm)/Au (30 nm)** **(liquid, floating gate)**	**Glass (borosilicate glass capillary)**	**FET at the end of a nanoneedle for intracell monitoring**	**[178]**
	MWCNT	PDMS	Non-covalent	25 µM	N/A (25 µM~5 mM (Not linear))	Capacitive measurement	Mold injection and thermal curing	MWCNT	PDMS	Measured at 2.4 kHz frequency	[179]
**O_2_**	**MWCNT**	**Au NP**	**Non-covalent**	**0.1 ppm**	**N/A** **(0~50 ppm)**	**Cyclic voltammetry**	**Not provided**	**Glassy carbon electrode**	**Glassy carbon**	**Not provided**	**[186]**
**Hydrogen peroxide**	**MWCNT**	**nitrogen doped Co-CNTs over graphene sheets**	**Non covalent**	**100nM** **3.4 ppb**	**−0.85 µA/ppm**	**Voltammetry, amperometry**	**Coating**	**Glassy carbon electrode**	**Glassy carbon electrode**	**No interference with uric acid, ascorbic acid and glucose**	**[188]**
	SWCNT	Phenyl capped aniline tetramer	Non covalent	<3 ppm	1%/ppm (3 ppm~8 ppm) Nonlinear <1%/100 ppm (48 ppm~1200 ppm)	Chemistor	Drop casting	Carbon ink	Glass	Not provided	[193]
**Free chlorine**	**MWCNT**	**Pristine**	**Non functionalized**	**<5 ppb** **(ClO^−^)**	**Logarithmic** **39%** **/decade *** **(** **0.03~8 ppm)**	**Chemistor**	**Dielectrophoresis** **(aligned MWCNT)**	**Cr/Au**	**Glass**	**No information about selectivity, pH information not provided**	**[191]**
	**MWCNT**	**Epoxy EpoTek H77A**	**Non covalent**	**20 ppb** **(HClO)**	**0.15 µA/ppb** **(** **0.02~4 ppm)**	**Voltammetry**	**Paste poured into tube and thermally cured**	**Epoxy/MWCNT composite**	**Not provided (tube)**	**Validated in real water matrices (tap water and swimming pool)**	**[194]**
	SWCNT	Phenyl capped aniline tetramer	Covalent	<60 ppb	92 nA/decade (0.06~60 ppm (linear up to 6 ppm))	Chemistor	Drop casting	Au	Glass	Non selective to different oxidants—list of oxidants not provided Regeneration possible	[53]
**Sulfite**	**MWCNT**	**Ferrocene**	**Non covalent (Physical immobilization)**	**0.1 µM**	**3.3 µA/µM** **(** **0.4 µM~4 µM** **)** **0.18 µA/µM** **(** **4 µM~120 µM)**	**Differential Pulse Voltammetry**	**Paste mixture with graphite powder blended with paraffin oil**	**MWCNT paste,** **Copper wire**	**Glass tube**	**Not provided**	**[78]**
**Sulfide**	**MWCNT**	**Hematoxylin**	**Non covalent**	**0.2 µM**	**103 nA/µM** **(0.5 µM~150 µM)**	**Amperometry**	**Paste mixture of MWCNT, mineral oil and graphite powder**	**Carbon paste**	**Teflon tube**	**No interference with Sn(II), Co(II), Pb(II), Zn(II), Cu(II), Ni(II), Mn(II), Fe(II) and Fe(III)**	**[197]**
	MWCNT	Pristine	Not functionalized	0.3 µM (CVD), 12.5 µM (ARC)	0.12 µA/µM (1.3 µM~113 µM) (CVD), 0.005 µA/µM (12.5 µM~87.5 µM) (ARC)	Hydrodynamic voltammetry	Drop casting	Glassy carbon electrode	Glassy carbon	Not provided	[199]
**Bisphenol A**	**SWCNT**	**βCD**	**Covalent**	**1.0 nM**	**1.3 mA/mM** **11 nM–19 µM**	**Cyclic voltammetry**	**Drop casting**	**Glassy carbon electrode**	**Glassy carbon**	**No interference study, but tested on real plastic samples**	**[206]**

Moreover, while the literature often claims that pristine or COOH-CNTs do not have selectivity, one observes here that non-functionalized CNT sensors with excellent limit of detections may operate free from interferents as well [110,129,147].

One may wonder whether the overall remarkably good performances of non-functionalized CNTs could be explained by an “effective” functionalization during the fabrication process. To clarify, additional molecules may remain (intentionally or not) on the CNTs sidewalls during device fabrication. Similarly, CNT synthesized in place by CVD may still carry leftover catalysts particles. The role of these by-products of fabrication is not addressed in the papers. A systematic study of the role of solvents and catalysts in the sensitivity to analytes in water could be valuable.

#### 4.2.3. Covalent versus Non-Covalent Functionalization

For pH, free chlorine, Cu(II), Hg(II) and Pb(II), covalent and non-covalent functionalization strategies are both reported. Except for Pb(II), non-covalent functionalization provides better performance than covalent functionalization. However, this conclusion should be tampered by the fact that it is never the same active compound being tested by both covalent or non-covalent functionalization. For instance, in the case of Pb(II) and Cd(II), bismuth is tested as an active compound of a lot of different functional probes, but all non-covalently functionalized. It would be very interesting to compare these results to a covalent functionalization strategy for bismuth or a bismuth derivative.

#### 4.2.4. On the Diversity of Functional Probes

The functional probes inventoried in this review cover a wide range of size scale and feature different levels of complexity. The literature includes primarily a large number of “single-component” functional probes (e.g., that can be described by a single chemical formula): mono or bi-atomic dopants; small or macromolecules; nanoparticles; polymers.

These materials can be coupled together, forming “composite probes”. Two-component strategies are fairly standardized now: a primary functional probe such as a polymer or a macromolecule is itself functionalized by a secondary probe (for instance PSS-Bismuth in [142]). Three (or more)-component strategies are also reported. For instance, in [151] PANI is functionalized with bismuth, and the resulting two-component functional probe is used to functionalize graphene oxide sheets.

Multi-component functional probes are thought to enhance the 3D structuration of the CNT layer, hence its specific surface area, and adsorption capability and thus its sensitivity. It is worth mentioning that these three-component structures often include flagship bidimensional materials such as graphene oxide and graphene.

These observations are confirmed in Table 17: among the 12 analytes where the best performance is achieved through functionalization, half of these are achieved through a multi-component strategy (four papers on two-component probes, two papers on three-component probes). Moreover, three papers out of 12 include graphene or graphene oxide and three papers out of 12 include a polymer (PANI or PSS) functionalized by a secondary probe.

#### 4.2.5. Type of CNT and CNT Alignment

The literature does not allow to compare between SWCNT against MWCNT as the choices between either is mostly guided by the transduction modes: MWCNTs are preferred for electrochemical and resistive sensing—as the CNT layer needs to be conducting; while the use of SWCNTs is mandatory with FETs. It would be very valuable to compare for the same analyte chemistors or electrodes with both types of MWCNTs (which is possible for electrochemical transduction and chemistors).

Similarly, there also appears to be no obviously optimal fabrication process. However, while definite comparison between references is not possible, it appears that alignment of CNT can provide outstanding performance, as is reported in [191] through dielectrophoresis or in [129] through threading.

### 4.3. Challenges and Perspectives

#### 4.3.1. Optimal Sensing and Sensing Mechanisms—The Role of Modelling

With only 90 references over 20 years covering eight categories of analytes and dozens of different functional probes, the literature remains too limited to derive unequivocally the best functionalization, transduction and design strategies for a given target analyte, as well as to propose mechanisms of sensing that are valid across the whole literature (for instance regarding the impact of sensor footprint and choice of contact materials).

To support research on that front, the larger availability of fine-grained room temperature operation models for functionalized CNT networks would be highly valuable. As of now, the multi-scale nature of these structures and the complexity of their room-temperature electrical and chemical behavior makes this a challenging proposition. The model for ohmic CNT networks proposed by Benda et al. [118] is a first step in this direction, but covers neither chemical effects nor field effects. Regarding chemical effects, studying the sensitivity in water of a functional probe toward target analyte is possible [209], but reliably including carbon nanotubes in the modelled system and further predicting electronic transport remains an unanswered challenge.

#### 4.3.2. Covering the Extreme Diversity of Analytes

This review details results on more than 25 target analytes. Despite this relatively large extent, and 20+ years of research, there are analytes of major impact that are not discussed by the literature, such as nitrate or the different forms of phosphate or iron in water. Furthermore, emerging contaminants such as drugs and hormones—which actually include dozens of relevant chemicals—will require monitoring tools in the near future. To cover more quickly the extreme diversity of analytes of interest, computer-aided design of functional probes should be promoted, as is reported in [209],

#### 4.3.3. Managing the Complexity of the Water Matrices through e-Tongue Strategy

By nature, water monitoring requires sensors that are robust to very complex water matrices, e.g., water that contains a large variety of interfering ions. This explains why a large number of the study reported here includes interferent studies. While some level of selectivity is often reported, it is rarely perfect with CNT-based sensors. Moreover, for obvious practical reasons, reports can rarely cover all the species that should be tested to account for the complexity real water matrices.

In the field of gaz sensing with CNT, which features the same issues, the solution that is widely used is the electronic nose [210]: an array of CNT sensors is fabricated, each CNT sensor carrying a specific functional probe or having different contact metals, in order to make it selective to a different specie. The CNT sensor array thus aims at multiparameter sensing. As selectivity is usually not achieved perfectly, machine-learning algorithms are applied to determine the footprint of the target gaz in the gaz matrix out of the multiple sensors data.

In the field fo sensing in liquids, the principle is called electronic tongue. Regularly used in CNT biosensors [211] and CNT sensors for food applications [212], it has so far remained mostly unexplored in the field of in-situ water monitoring, While recent work has shown the posibility to actually fabricate sensor arrays compatible with water monitoring application [213,214], a large amount of work remains to be done to prove the reliability of multiparameter measurements in complex water matrices.

#### 4.3.4. Toward Real Applications: The Need for Ageing Studies

Besides the challenge of reliability measurements, water quality monitoring requires survival of the sensors in water over relatively long period of time, preferably without calibration. A chemical sensor with one year of operation in the field without recalibration would be a game changer. By contrast, ageing studies included in references so far discuss only short term cyclability (5 to 10 cycles of measurements) and short term sensor survival (usually a couple of days); the change in sensitivity over time is hardly studied. There is a major discrepancy between what is currently done in laboratory in terms of ageing and what is required by end users for true applications. Major progress in marketability of CNT sensors for water applications could be expected if standardized protocols for ageing studies could be developped with reasonsable scope and duration and then generalized among research teams.

## 5. Conclusions

This review identifies and compares 90 CNT-based water quality monitoring sensors reported from 2000 to April 2021. A set of 126 additional references provide context and supporting information. After reviewing the challenges of on-line drink water quality monitoring and presenting the highlights of CNT-based electronic devices and electrical transduction for chemical sensing, a quantitative comparison of the performances of reported sensors based on limit of detection, sensitivity and detection range was proposed. The target analytes are pH, micronutrients and heavy metals, nitrogen forms, sulfur forms, disinfectants and dissolved oxygen, as well as miscellaneous materials relevant to drink water quality.

Overall, there are so many parameters featuring in the design and operation of CNT based water quality sensors that a systematic comparison across all of these references was not possible with the current extent of the literature. However, some key conclusions can be drawn regarding the best transduction mode and functional probe considering a specific analyte.

Across all analytes, while electrochemical sensing with MWCNTs is the most frequently reported approach and allows to reach remarkable limit of detection (down to the ppt level), FET and chemistor approaches—which are much less frequently used—may also reach detection limits in the ppt range. In the rare instances where they are tested for the same analyte, FETs may perform as well or better than MWCNT electrodes, while chemistors usually perform worse than both FETs and electrochemical sensors. Overall, a more extensive evaluation of FET and chemistors for various analytes would be valuable.

A large variety of functional probes is reported. They cover the full-size scale from single atomic dopants to polymers and often couple two to three chemical building blocks. While these probes provide remarkable performances, especially the multi-component ones, there are—surprisingly—several analytes for which non-functionalized or COOH-functionalized CNTs provide better performances (pH, Zn(II), free chlorine). Non-functionalized or COOH-functionalized CNTs sensors are also reported to allow selectivity and to be resilient to interferents. These results suggest to systematically compare in new studies the performances of functionalized and non-functionalized CNTs.

Finally, despite the large numbers of analytes covered here, there are still a lot of highly environmentally relevant analytes that are not covered at all by the literature, for instance the various forms of iron (Fe(II), Fe(III)), ammonium, nitrate and phosphate ions or the various pesticides found at a ppb level in drink water. In addition, the literature does not address emerging contaminants such as drugs (e.g., antibiotics, hormones) and their degradation by-products or biological materials such as bacteria. While a few of these targets may be covered by the literature about optical CNT sensors, these results do not transfer easily to online monitoring applications, so that there are still a lot of research opportunities for electrical and electrochemical CNT sensors.

## Figures and Tables

**Figure 5 sensors-22-00218-f005:**
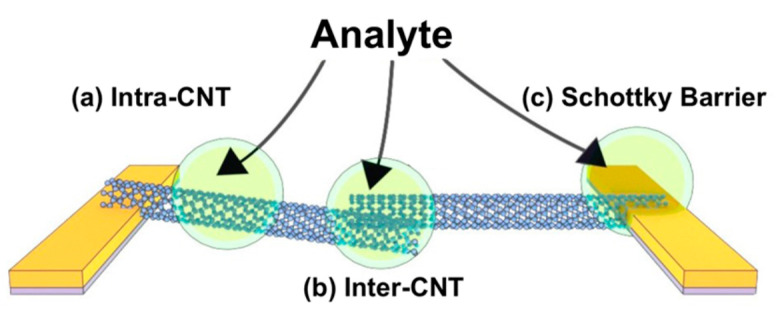
Schematic of possible sensitivity sites which affect the conductivity: (**a**) at the sidewall or along the length of the CNT itself, (**b**) interface between CNT-CNT (inter-CNT) and (**c**) at the interface between the metal electrodes and the CNT (Schottky barrier). Reprinted with permission from [15]. Copyright (2018) from American Chemical Society.

**Figure 6 sensors-22-00218-f006:**
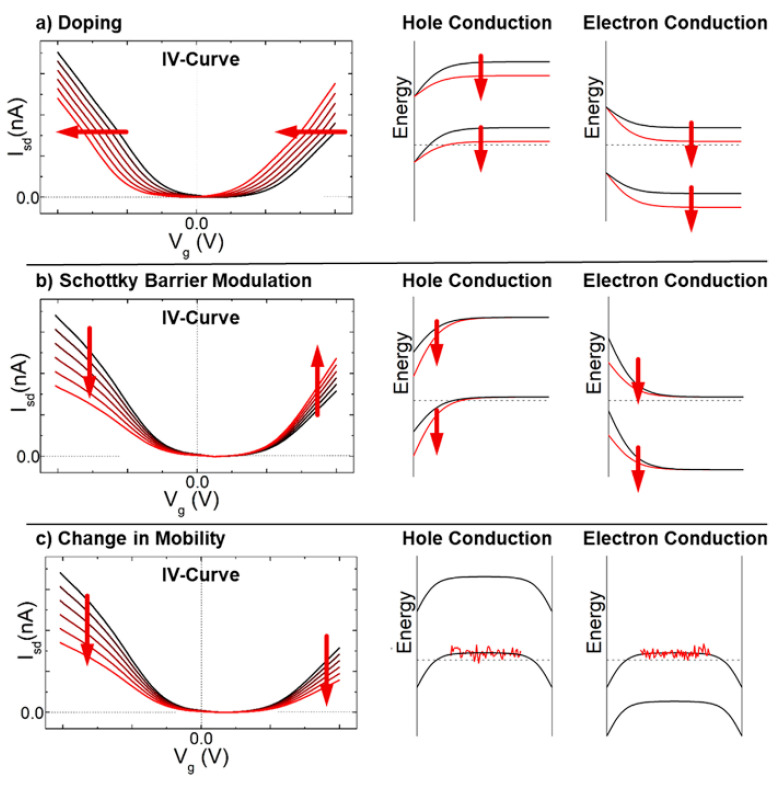
Theoretical I-V curves of a chemFET depending on (**a**) the doping (**b**) Scottky barrier modulation (**c**) change in mobility. Reprinted with permission from [15]. Copyright (2018) from American Chemical Society.

**Figure 7 sensors-22-00218-f007:**
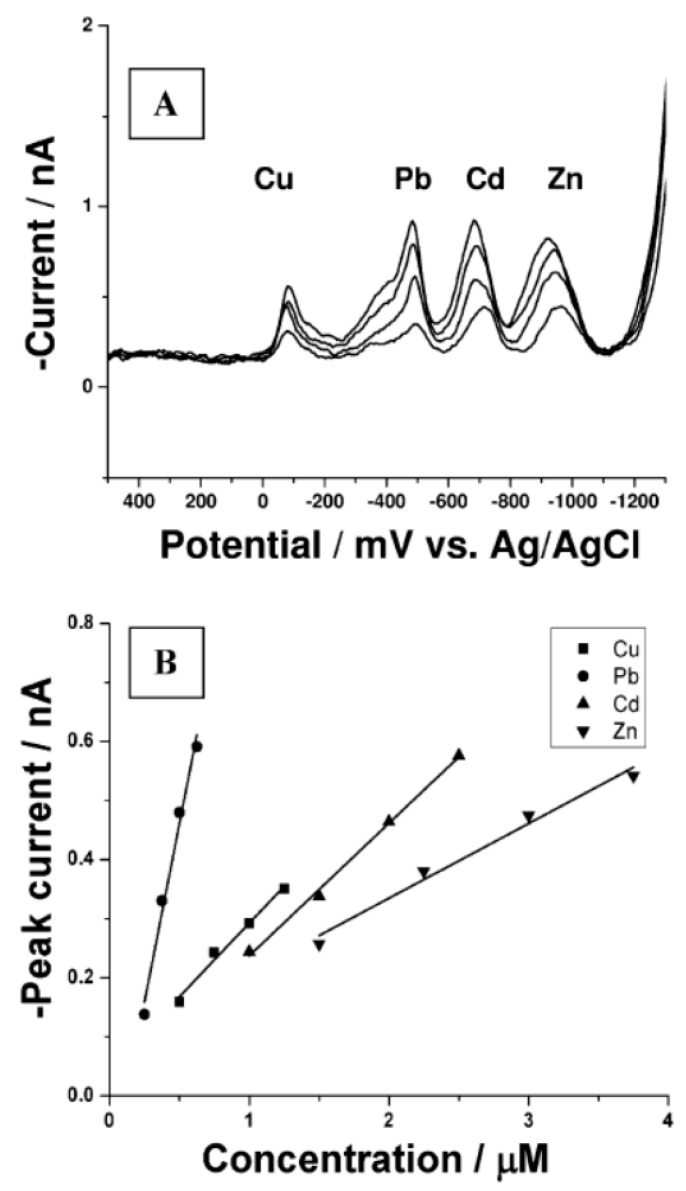
(**A**) Simultaneous detection of Cu(II), Pb(II), Cd(II) and Zn(II), ion concentrations 0.5, 0.25, 1.0, 1.5 µM for Cu(II), Pb(II), Cd(II), Zn(II), respectively; and 1.5, 2, 2.5 times of above concentrations for these metals ions. (**B**) Calibration Curve for Cu(II), Pb(II), Cd(II), Zn(II). Accumulation time: 120 s, deposit potential: −1.5 V. Reproduced from [129].

**Table 1 sensors-22-00218-t001:** Overview of reported CNT-based water quality monitoring sensors.

Type of Analytes		Numbers of Refs.	Ref. SWCNT	Ref. MWCNT	Ref. CNTFET	Ref. Chemistors	Ref. EC	Ref. Functionalized (COOH Excluded)
**All Analytes**		**90**	**26 (29%)**	**64** **(71%)**	**11** **(12%)**	**13** **(14%)**	**66** **(73%)**	**74** **(82%)**
pH		16 (18%)	12	4	6	7	5 (2 with CNTFET)	8
Micronutrients and toxic metals (total)	All included	36 (40%)	5	32	2	2	33	31
	Pb(II)	16 (18%)	0	16	0	0	16	12
	Cd (II)	13 (14%)	0	13	0	0	13	11
	Cu(II)	9 (10%)	1	9	1	0	8	8
	Hg(II)	8 (9%)	3	5	1	1	6	6
	As(III)	5 (6%)	0	5	0	0	5	5
	Zn(II)	4 (4%)	0	4	0	0	4	2
	Miscellaneous	2 (2%)	2	0	1	1	0	2
Nitrite		10 (11%)	1	9	0	0	10	10
Water hardness		2 (2%)	1	1	1	0	1	2
DO		2 (2%)	0	2	0	0	2	2
Disinfectants	Free chlorine	3 (3%)	1	2	0	2	1	2
	Hydrogen peroxide	6 (7%)	1	5	0	1	5	6
Sulfur	Sulfide	4 (4%)	0	4	0	0	4	4
	Sulfite	2 (2%)	0	2	0	0	2	2
Miscellaneous		9 (10%)	4	5	2	1	5	8

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
