# Peer review of "Electrical and Electrochemical Sensors Based on Carbon Nanotubes for the Monitoring of Chemicals in Water—A Review"

_sensors, 2021, doi:10.3390/s22010218_

Round 1

Reviewer 1 Report

I am attaching my comments.

Author Response

  1. The statement “So far there is however no review focusing specifically on drinking water quality monitoring based on carbon nanotube sensors, a gap which the present paper endeavors to ” is not understandable.

The sentence has been replaced as follows

However, there has been no review focusing specifically on water quality monitoring based on carbon nanotube sensors . The present paper endeavors to fill this gap.

  1. “After summarizing the chemical parameters of interest regarding drink water quality, we present operating principles of carbon nanotube- based chemical sensors, including a description of the three electrical transduction modes, a description of the different fabrication strategies” should be clearly

The sentence has been shortened as follows:

After a brief overview of the design, fabrication and operating principles of CNT-based chemical sensors,

The review contains a large number of electrochemical determination of metal ions that would not fall into this descriptions.

The formulation has been replaced by “electrical and electrochemical” as in the title

Moreover the authors do not discuss the  techniques such as stripping voltammetry, cyclic voltammetry etc.

This is discussed in section 2.3.2. Stripping voltammetry is described page 8, line 309. Cyclic voltammetry was described but the usual name not provided. It is now corrected

There are different types of voltammetry depending on the way the voltage is applied, notably linear sweep or pulse-wise increase. The latter (usually called differential pulse voltammetry), is reported to be well suited for solid electrodes based on organic compound and more sensitive than the former (usually called voltammetry)

  1. Table 1 shows comparison between liquid-phase deposition methods for It should include references.

References have been added

  1. The authors state” The second type of processes are called non-faradaic. They include processes of desorption and adsorption occurring at the electrode-electro- lyte interface which may impact the electrical response of the Pro-cesses of ionic transport within the electrolyte are also determining, as they enable the movement of charges between the electrodes required for faradaic processes to occur”. Non-faradaic process by normal usage refers to charging currents. Here the authors use it for adsorption and desorption processes. It is well known that adsorbed species in several situations can undergo electron transfer reactions.

Thank you for pointing out this mistake. The sentence has been corrected and shortened as

Two types of process may occur in electrochemical cells. In a Faradaic process, charge particles transfer from electrode to electrode through the electrolyte. In non-Faradaic, charge is progressively stored [1].

  1. The authors state “Electrochemical impedance spectroscopy (EIS) is a derivative of volt ammetry where the voltage is alternating (AC) and applied at different ” This is confusing and should be properly described,

The sentence has been clarified as follows:

In electrochemical impedance spectroscopy (EIS), an alternating voltage is applied. The phase shift and amplitude of the current are measured over a range of frequencies. It provides information on the rate of the electrochemical reactions and on the ionic transport in the electrolyte [62].

  1. Electrochemical impedancemetry (Nyquist plot: real impedance vs imaginary impedance for different of electrodes). Reproduced from [65].  Nyquist figure shows a,b and

They  should be explained in the caption.

Legend has been modified as follows:

Electrochemical impedance spectroscopy (Nyquist plot: real impedance vs imaginary impedance for different of electrodes; a – Bare glassy carbon electrode (GCE) 1; b – CNTs/poly(1,2-diaminobenzene) prepared by cyclic voltammetry modified GCE ; c – CNTs/poly(1,2-diaminobenzene) composite prepared by multipulse potentiostatic method modified GCE).

  1. “There are four main types of device architecture for CNT-FET chem ical sensors: top gate, bottom gate, liquid gate and hybrid structures, with respectively 3, 1, 5 and 2 papers out of 11”  what are the numbers and what is 2 papers out of 11?

The text has been modified as follows:

Among 11 reported CNT-FET-based chemical sensors, 2 are top-gated Figure 4).

  1. “Table 3 contains irrelevant pH sensors for water .” It is  stated in the beginning of this section recommended levels of pH for drinking water are 5 to 8.5 by WHO. Table contains a large number of other pH values. Is that necessary?

Yes in practice because it is expected from most commercial pH sensors for water monitoring to cover a larger range of pH. The following sentence has been added:

Commercial pH sensors for water monitoring often cover larger ranges of pH (regularly from 1 to 14) to address abnormal situations regarding drink water (e.g. pollutant ingress) and to be applicable to industrial water processing (chlorination, waste water, aquaculture) as well.

  1. It is stated in conclusions “This review addresses advances of CNT-based water quality monitoring sensors reported from 2000 to April 2021 across 90 references”. The manuscript contains 200

The sentence has been modified as follows:

This review identifies and compares 90 CNT-based water quality monitoring sensors reported from 2000 to April 2021. A set of 126 additional references provide context and supporting information.

  1. The authors have given a reference 59 as
    1. Bard and L. R. Faulkner, "Electrochemical Methods: Fundamentals and Applications," Russ. J. Electrochem, vol. 38, p. 1364–1365, 2001. This should be corrected as A.J. Bard and L.R. Faulkner, “Electrochemical Methods: Fundamentals and Applications,” Wiley, NJ (2001).

The reference has been corrected

  1. While the authors have done a very good job of examining carbon nanotubes for a specific application, it has too many diversions from drinking I would suggest the authors to condense the material for the specific application without distractions.

Thank you.

It is correct that some reports in this review provide performances beyond those needed for drinking water only, for instance in terms of limit and ranges of detection. However, from a research perspective, it allows to assess the technical  limits of the approaches (in terms of best performances). Moreover, in the prospects of turning the research work into commercial products, end-users often require wider range of operation than what is purely requested by regulations, both to deal with abnormal situations and to be able to use the products in various water-related applications beyond drink water only.  

Reviewer 2 Report

The subject is vast and the undertaking is huge. I'm glad someone had the courage to do it.

Obs.1. Chapter 2.2.3. Recent investigations and achievements state that chemical vapor deposition processes also occur at temperatures below 750-9500C.

Obs.2. Chapter 2.2.3. Table 1. Usually drop-casting method is followed by dielectrophoretic alignment so the quality of the final product, using this succession, can be better than drop-casting alone product. The authors must introduce a comment showing that these methods, although very good individually, can also be used in a combined way.

Obs.3. I recommend to the authors to insist and also present issues related to the presence of hysteresis, caused by water molecules in carbon nanotube field-effect transistors for example.

Obs.4. Chapter 3.2.3. In the case of Zinc (II) due to the fact that authors used four references the conclusion that pristine MWCNTs have a remarkably better limit of detection that functionalized MWCNTs (with bismuth) can be debated. Reference [120] contradicts authors conclusion.

Obs.5. I suggest to the authors to think about the challenges related to the detection of antibiotics or hormones although the subject is partially mentioned in the last chapter. The most interesting topic about pesticides is the historical accumulation.

Author Response

Obs.1. Chapter 2.2.3. Recent investigations and achievements state that chemical vapor deposition processes also occur at temperatures below 750-9500C.

A more recent reference has been provided to cover low-temperature growth of CNT and the sentence has been modified as follows:

CNTs are usually synthesized via chemical vapor deposition (CVD) directly onto pre-patterned electrodes within a temperature range from 550°C to 1000°C [46, 47].

Obs.2. Chapter 2.2.3. Table 1. Usually drop-casting method is followed by dielectrophoretic alignment so the quality of the final product, using this succession, can be better than drop-casting alone product. The authors must introduce a comment showing that these methods, although very good individually, can also be used in a combined way.

Indeed, the formulation on dielectrophoresis and the table are unclear on the topic. DEP requires deposition of a liquid CNT suspension on a substrate as a first step; deposition which is very often achieved by drop casting.  

The sentence on DEP has been modified as follows:

After substrate deposition of a liquid CNT suspension, particularly fol-lowing drop casting, dielectrophoresis may then be used to improve on the deposition quality, notably to control accurately CNT positioning or to achieve CNT alignment [60].

The table has been removed to shorten the review.  

Obs.3. I recommend to the authors to insist and also present issues related to the presence of hysteresis, caused by water molecules in carbon nanotube field-effect transistors for example.

The following sentence has been added with complementary citations

Regarding the latter, this hysteresis is attributed in large part to the adsorption of water molecules on the device surface creating charging effect [2]. Hence, it is expected to be a particularly relevant indicator in CNTFET-based water quality sensors but there has no systematic study on it so far [3].

Obs.4. Chapter 3.2.3. In the case of Zinc (II) due to the fact that authors used four references the conclusion that pristine MWCNTs have a remarkably better limit of detection that functionalized MWCNTs (with bismuth) can be debated. Reference [120] contradicts authors conclusion.

The reviewer is correct that NOT ALL pristine MWCNT have better limit of detection than functionalized MWCNT and the formulation was inaccurate. The paragraph has been modified as follows:

Unlike results on lead and cadmium where functionalized CNTs have much better performances than non-functionalized CNTs, one observes here that one of the two pristine MWCNTs references [119] has a re-markably better limit of detection (0.09 ppb – two orders of magnitude lower) than the two references with MWCNTs non covalently func-tionalized with bismuth. By contrast, the other reference with pristine MWCNT has worse detection limit the Bi-functionalized devices. It points out once again the strong sensitivity of water quality sensor performances to CNT processing (CNT thread versus CNT paste) and to transduction mode (stripping voltammetry versus potentiometry).

Obs.5. I suggest to the authors to think about the challenges related to the detection of antibiotics or hormones although the subject is partially mentioned in the last chapter. The most interesting topic about pesticides is the historical accumulation.

This is now discussed in section 4.3. Perspectives and challenges.

Reviewer 3 Report

The authors present an extensive review on electrochemical and resistive sensors based on carbon nanotubes for detecting chemical in water. It contains a lot of literature and many tables for summarizing the report. I do strongly recommend the publication in the journal.

A minor comment: some of references deal with just water, not for drink water. Could you remove "drink" in the title? 

Author Response

The authors present an extensive review on electrochemical and resistive sensors based on carbon nanotubes for detecting chemical in water. It contains a lot of literature and many tables for summarizing the report. I do strongly recommend the publication in the journal.

We thank the reviewer for the kind comment.

A minor comment: some of references deal with just water, not for drink water. Could you remove "drink" in the title? 

It is discussed and changed.

Reviewer 4 Report

The manuscript entitled “Electrical and electrochemical sensors based on carbon nanotubes for the monitoring of chemicals in drink water - A review” is an detailed review article about CNT usage as an active material for drinking water purity monitoring. The article consist of introduction about the  necessity of the monitoring the quality of the drinking water as a global issue. The authors address the challenges and motivation of that research area. Next the authors present the types of the sensor architectures of the sensor and their principles of operations commonly used in the water quality monitoring area. After the device description paragraph authors briefly introduce the measured parameters , pH or the contend of different elements in drinking water, followed by detailed discussion of each parameter or element described in the literature.

 The manuscript is ended with brief summary, conclusions and discussion about future development of this branch of the CNT based sensor applications.

In my opinion the review article is interesting for the reader, with brief comparison of the most important results for selected analytes. The additional advantage of the article is table system with summary of the research related with describe of sensor type, analyte type, LOD and interferences.

My major concern is with the length of the review. Maybe the authors will consider reduction of the sensor principle paragraph, it is helpful for the reader to know the principle of operation of the described sensor types but it is not the main aim of the review article.

Please find my comments in points:

Point 1: The authors should provide legend to table 1 starting in line 249. Marks + and – are intuitive but it would be nice for the readers to see what ++ and +++ description means.

Point 2: Authors should look over the quotation system for instance in line 363 they use comma to separate the references but in lines for instance 351, 1262, 1319 they just add references after space.

Point 3: Figure descriptions should be A,B,C or 1,2,3 not left- right like in for instance figure 4. It will help the reader to follow the references to the figures in text.

Point 4: Authors use interchangeably chemical formula for elements and their names for instance in table 2 (names of the elements) line 730 formula, paragraph 4.1.1. names (line 1409 Zinc line 1430 formula).

Point 5: The authors should check the tables editing. The table from page 74 in column type of CNT SWCNT is written together but MWCNT is separated.  

Author Response

 Thank you for the kind comment.

My major concern is with the length of the review. Maybe the authors will consider reduction of the sensor principle paragraph, it is helpful for the reader to know the principle of operation of the described sensor types but it is not the main aim of the review article.

We considered transforming the entirety chapter 2 into supplementary material, but the present journal is a multi-disciplinary review so a significant range of researchers may not know the basics of CNT sensors and of chemical sensors. So we left the necessary minimum to highlight all the major topics of concern regarding the CNT sensors discussed further in the text. To reduce further, we removed Table 1 and Figure 3.

Please find my comments in points:

Point 1: The authors should provide legend to table 1 starting in line 249. Marks + and – are intuitive but it would be nice for the readers to see what ++ and +++ description means.

 The table is slightly beside the scope of the present paper and has been removed to reduce the length of the paper.

Point 2: Authors should look over the quotation system for instance in line 363 they use comma to separate the references but in lines for instance 351, 1262, 1319 they just add references after space.

with single quotation system.

For example: From [1],[2] to [1,2]

Point 3: Figure descriptions should be A,B,C or 1,2,3 not left- right like in for instance figure 4. It will help the reader to follow the references to the figures in text.

  It has been corrected with correctly indicating the figures, for example, A, B or C.

Point 4: Authors use interchangeably chemical formula for elements and their names for instance in table 2 (names of the elements) line 730 formula, paragraph 4.1.1. names (line 1409 Zinc line 1430 formula).

   It has been corrected by changing the formula.

 For example: from Zn2+ to Zn (II), or As3+ to As(III)

Point 5: The authors should check the tables editing. The table from page 74 in column type of CNT SWCNT is written together but MWCNT is separated.  

We considered sorting the references based on the type of transduction methods. For example in page 74 (which is page 71 in the latest version), type of transduction method (Chemistor, transistor or electrochemical sensors) is the first considered factor and the second factor was the type of CNTs.

Reviewer 5 Report

The article gives an overview of electrical and electrochemical sensors based on carbon nanotubes for monitoring water quality and pollutants.
The search for drinking water is one of the most important challenges that mankind is facing, so the topic of the paper is extremely interesting.

The article is detailed and covers the subject exhaustively, filling a gap in the literature about carbon nanotube sensors.

The manuscript can be suitable for publication after a minor revision:

1. Lines 407,437,454: I recommend using the same notation used in the caption to reference the figures in the text (example figure 4 (a), etc.).

  1. Line 508: “In general, the CNT-FET channel may be formed either by a single semi-conducting 509 SWCNT or by a percolating network of SWCNTs with a semi-conducting behavior.” Examples of CNT-FET with a percolating network of SWCNTs is studied in https://doi.org/10.1088/0957-4484/21/11/115204, that could be added as reference to support this sentence.

  2. Line 888: 6 MWCNTs (“the last 6 reports being with functionalized MWCNT”) are mentioned in the text, while only 5 are listed in table 7.

  3. Section 3.2.6 shows 7 papers, but the table shows 9 and two are missing references.

  4. To avoid that the paper is a simple list of data reported in literature, I suggest adding a subsection in the discussion where the authors give their personal perspective on the technical development and the market impact of the discussed sensors.
  5. Buckypaper fabricated from multi-walled carbon nanotubes has been used to remove pollutants from water and improve the quality of drinking water. The authors could check if there are any buckypaper based sensors to report in their review.

Author Response

Thank you for your kind comments

  1. Lines 407,437,454: I recommend using the same notation used in the caption to reference the figures in the text (example figure 4 (a), etc.).

by using the same notation as reviewer suggested, for example figure 4(a)… etc.

  1. Line 508: “In general, the CNT-FET channel may be formed either by a single semi-conducting 509 SWCNT or by a percolating network of SWCNTs with a semi-conducting behavior.” Examples of CNT-FET with a percolating network of SWCNTs is studied in https://doi.org/10.1088/0957-4484/21/11/115204, that could be added as reference to support this sentence.

  1. Line 888: 6 MWCNTs (“the last 6 reports being with functionalized MWCNT”) are mentioned in the text, while only 5 are listed in table 7.

It has been corrected

  1. Section 3.2.6 shows 7 papers, but the table shows 9 and two are missing references.

This is now corrected

  1. To avoid that the paper is a simple list of data reported in literature, I suggest adding a subsection in the discussion where the authors give their personal perspective on the technical development and the market impact of the discussed sensors.

This is now discussed in section 4.3. Perspectives and challenges.

  1. Buckypaper fabricated from multi-walled carbon nanotubes has been used to remove pollutants from water and improve the quality of drinking water. The authors could check if there are any buckypaper based sensors to report in their review.

Buckypaper designates a thin, freestanding film made out of CNT network, often achieved through vacuum filtering and often of macroscopic dimensions. We found no reference with the exact wording “buckypaper”, as the specific geometry which makes buckypaper so valuable for other applications (water purification, strain sensing) is not of particular advantage for water monitoring. However, the following reference on pH sensing is actually buckypaper, though the material is left on the filter and not fully freestanding.

  1. Stojanović, T. Kojić, M. Radovanović, D. Vasiljević, S. Panić, V. Srdić and J. Cvejić, "Flexible sensors based on two conductive electrodes and MWCNTs coating for efficient pH value measurement," J. Alloys Compd, vol. 794, p. 76–83, 2019.

The text has been modified accordingly: sensor fabrication based on vacuum filtering is now mentioned as well the concept of buckypaper.

Round 2

Reviewer 5 Report

The authors have considered all the reviewers' suggestions. Their replies and corrections to the paper are satisfactory.